# Structural Characterization of Titanium–Silica Oxide Using Synchrotron Radiation X-ray Absorption Spectroscopy

**DOI:** 10.3390/polym14132729

**Published:** 2022-07-03

**Authors:** Arpaporn Teamsinsungvon, Chaiwat Ruksakulpiwat, Penphitcha Amonpattaratkit, Yupaporn Ruksakulpiwat

**Affiliations:** 1School of Polymer Engineering, Institute of Engineering, Suranaree University of Technology, Nakhon Ratchasima 30000, Thailand; arpaporn.te@gmail.com (A.T.); charuk@sut.ac.th (C.R.); 2Center of Excellence on Petrochemical and Materials Technology, Chulalongkorn University, Bangkok 10330, Thailand; 3Research Center for Biocomposite Materials for Medical Industry and Agricultural and Food Industry, Nakhon Ratchasima 30000, Thailand; 4Synchrotron Light Research Institute (SLRI), 111 University Avenue, Muang District, Nakhon Ratchasima 30000, Thailand; penphitcha@slri.or.th

**Keywords:** mixed oxide, sol–gel method, titania, silica, XAS

## Abstract

In this study, titania–silica oxides (Ti_x_Si_y_ oxides) were successfully prepared via the sol–gel technique. The Ti and Si precursors were titanium (IV), isopropoxide (TTIP), and tetraethylorthosilicate (TEOS), respectively. In this work, the effects of pH and the Ti/Si atomic ratio of titanium–silicon binary oxide (Ti_x_Si_y_) on the structural characteristics of Ti_x_Si_y_ oxide are reported. ^29^Si solid-state NMR and FTIR were used to validate the chemical structure of Ti_x_Si_y_ oxide. The structural characteristics of Ti_x_Si_y_ oxide were investigated using X-ray diffraction, XRF, Fe-SEM, diffraction particle size analysis, and nitrogen adsorption measurements. By applying X-ray absorption spectroscopy (XAS) obtained from synchrotron light sources, the qualitative characterization of the Ti–O–Si and Ti–O–Ti bonds in Ti–Si oxides was proposed. Some Si atoms in the SiO_2_ network were replaced by Ti atoms, suggesting that Si–O–Ti bonds were formed as a result of the synthesis accomplished using the sol–gel technique described in this article. Upon increasing the pH to alkaline conditions (pH 9.0 and 10.0), the nanoparticles acquired a more spherical shape, and their size distribution became more uniform, resulting in an acceptable nanostructure. Ti_x_Si_y_ oxide nanoparticles were largely spherical in shape, and agglomeration was minimized. However, the Ti_50_Si_50_ oxide particles at pH 10.0 become nano-sized and agglomerated. The presence of a significant pre-edge feature in the spectra of Ti_50_Si_50_ oxide samples implied that a higher fraction of Ti atoms occupied tetrahedral symmetry locations, as predicted in samples where Ti directly substituted Si. The proportion of Ti atoms in a tetrahedral environment agreed with the value of 1.83 given for the Ti–O bond distance in Ti_x_Si_y_ oxides produced at pH 9.0 using extended X-ray absorption fine structure (EXAFS) analysis. Photocatalysis was improved by adding 3% wt TiO_2_, SiO_2_, and Ti_x_Si_y_ oxide to the PLA film matrix. TiO_2_ was more effective than Ti_50_Si_50_ pH 9.0, Ti_50_Si_50_ pH 10.0, Ti_50_Si_50_ pH 8.0, and SiO_2_ in degrading methylene blue (MB). The most effective method to degrade MB was TiO_2_ > Ti_70_Si_30_ > Ti_50_Si_50_ > Ti_40_Si_60_ > SiO_2_. Under these conditions, PLA/Ti_70_Si_30_ improved the effectiveness of the photocatalytic activity of PLA.

## 1. Introduction

In recent years, binary oxides of Ti and Si have gained a great deal of attention in a wide range of applications, including TiO_2_–SiO_2_ [1], Ag–TiO_2_–SiO_2_ [2], and the ordered mesoporous TiO_2_/SBA-15 matrix [3]. Sol–gel processing is an adjustable method that has attained commercial success because it allows for the control of the desired properties of a material during the preparation process. As a result, the sol–gel method has become one of the most preferred methods to produce oxide materials. Titanium dioxide (TiO_2_), often known as titania, is an important substance. This multifunctional material may be used in coatings [4,5], biomaterials [6], photocatalysts [7], food packaging [8], and chemical sensors [9], to mention a few applications. It possesses a number of desirable characteristics, including chemical resistance, chemical stability [10], photocatalytic activity, outstanding photostability, biocompatibility, and antibacterial activity [11]. Furthermore, titanium dioxide (TiO_2_) is widely recognized as a large-bandgap semiconductor with photocatalytic activity, whereas silicon dioxide (SiO_2_), often known as silica, has a well-defined ordered structure, a large surface area, is cost-effective to produce, and is easy to modify [12]. 

Adding SiO_2_ to the formulation of a polymer is also known to increase its modulus of elasticity, strength, heat and fire resistance, wear resistance, insulating properties, and other properties [13]. In addition to drug delivery, bioimaging, gene transport, and engineering, SiO_2_ is employed as a food ingredient. Moreover, the FDA has classified silica as a “generally recognized as safe” (GRAS) substance, making it an attractive candidate for biological activities [14].

Titanium–silicon oxide (Ti_x_Si_y_) is a fascinating material family that has gained a great deal of attention in recent years. The oxide has been widely employed as a catalyst and support for a variety of processes. It is constructed of TiO_2_ and SiO_2_, and it not only possesses the advantages of both TiO_2_ (photocatalytic properties and antimicrobial biomaterials) and SiO_2_ (high thermal stability and excellent mechanical strength), but it also expands their applications by developing additional catalytic active sites, owing to the chemical link between the two materials [15]. There are several techniques for manufacturing Ti_x_Si_y_ oxide, but sol–gel approaches appear to be the most cost-effective. These methods do not necessitate the use of costly solvents. Titanium alkoxide (e.g., titanium isopropoxide) is frequently utilized in sol–gel techniques.

One of the most prevalent methods for producing oxide materials is the sol–gel method. This method allows for the management of textural characteristics, as well as a high degree of mixing in bulk mixed oxides while retaining the oxide material anion-free. Furthermore, the materials produced using this method are not microstructurally ordered, have a large surface area, and show pore size variation [16,17]. 

The powerful technique of synchrotron X-ray absorption spectroscopy (XAS) using tunable, very intense X-rays from a high-energy electron storage ring has been applied to investigate the structural properties of materials. XAS is particularly attractive because of its ability to deliver electronic structure as well as geometric information. A typical *K*-edge absorption spectrum is divided into two sections: (i) the X-ray absorption near-edge structure (XANES) (<50 eV) contains bond lengths and angles, as well as information about the three-dimensional structure around the absorbing atom; (ii) the extended X-ray absorption fine structure (EXAFS) region (typically > 50 eV) provides information on the initial coordination shell around the absorbing atom, including coordination numbers and bond lengths [18].

In most of the previous reports, XAS has been employed to determine information on the coordination environment of tetravalent Ti[Ti(IV)] in structurally complex oxide materials and bond distances between Ti–O and Ti–Si atoms. Niltharach et al. [19] used XANES methods to investigate the structural details of sol–gel-produced TiO_2_ samples with and without the inclusion of Ce. The XANES results also showed that the sample produced under the low hydrolysis condition had a significant number of Ti atoms in forms other than anatase and rutile TiO_2_. Won Bae Kim et al. [20] used the linear combination of two reference XANES spectra to estimate the pre-edge of the Ti *K*-edge in order to quantitatively analyze the percentages of Ti–O–Si and Ti–O–Ti bonds. The findings of pre-edge fitting in conjunction with XRD and XPS suggested that monolayer coverage was attained at around 7–10 wt.% Ti loading, where the concentration of Ti in Ti–O–Si was saturated to 0.56 mmol-Ti/g material. Shuji Matsuo et al. [21] determined the local Ti environments in the sol, gel, and xerogels of titanium oxide prepared by a sol–gel method using titanium *K*-edge XANES. All of the samples could be divided into three groups: the anatase group, the anatase-like structure group, and the weak Ti–Ti interaction group.

A variety of materials were produced by changing the composition of the primary sol and/or the thermal treatment. The nature of the precursor, the molar ratio between silicon and titanium alkoxide, the type of solvent, and the use of modifying agents all had an influence on the final material’s microstructure and hence its properties. Furthermore, while manufacturing Ti_x_Si_y_ oxide via sol–gel, the pH of the solution is one of the most essential factors determining Ti_x_Si_y_ oxide’s properties. The hydrolysis and condensation behavior of a solution during gel formation, as well as the structure of the binary oxide, is influenced by its pH [22]. In order to maintain the sample in a powder morphology with a large surface area, the gel must be stabilized. The goal of this study was to evaluate the effects of pH and Ti/Si atomic ratio on TixSiy oxide using the sol–gel technique (Stöber method). XANES and EXAFS were used to evaluate Ti–O–Si and Ti–O–Ti connectivity in Ti_x_Si_y_ oxides, the local atomic structure, the bond distances between Ti–O and Ti–Si atoms, the coordination number, and the valence state of titanium atoms TiO_2_ and all mixed oxide samples. In addition, the photocatalytic behavior of all samples was investigated by comparing their degradation of methylene blue (MB) solutions exposed to UV radiation.

## 2. Materials and Methods

### 2.1. Materials

Tetraethylorthosilicate (TEOS, 98%, AR-grade) and titanium (IV) isopropoxide (TTIP, 98%, AR-grade) were purchased from Acros (Geel, Belgium). Absolute ethanol (C_2_H_5_OH, AR-grade), hydrochloric acid (HCl, AR-grade), and ammonium hydroxide (NH_4_OH, AR-grade) were supplied by Carlo Erba Reagents (Emmendingen, Germany). PLA4043D was provided by NatureWorks LLC (Minnetonka, MN, USA).

### 2.2. Synthesis of Titania–Silica Binary Oxide (Ti_x_Si_y_ Oxide), Silica, and Titanium Dioxide

Sol–gel techniques were used to prepare titanium–silicon oxide (Ti_x_Si_y_) utilizing a modified Stöber procedure, including the simultaneous hydrolysis and condensation of TEOS [20]. Figure 1 illustrates the experimental technique for producing Ti_x_Si_y_ oxide particles. TEOS and TTIP were used without purification. 

The total moles of Ti-alkoxide and Si-alkoxide were determined to be 0.12 mol. For the pre-hydrolysis step, the mixture of two alkoxides was added into a five-neck round-bottom flask containing 150 cm^3^ of ethanol for 1 h. A condenser was used in the condensation step to prevent the solvent from vaporizing out of the process. Stirring raised the temperature of the reactor to 75–80 °C. After an hour of mixing, a yellowish translucent sol was observed after adding TTIP and 25 ml of ethanol to the reactor drop by drop for 1 h. DI water was added to the sol, which was then aged at room temperature for 2 h at 80 °C. The aged gel was then refined in a centrifugal separator for 10 min at 6000 rpm. Finally, the solid particles were calcined for 3 h at 450 °C. A reference pure SiO_2_ sample was prepared in NH_4_OH in the same manner as described above but without the addition of TTIP [23]. In addition, reference pure TiO_2_ was generated in the same maner as Ti_x_Si_y_ oxide in acid [24]. Table 1 and Table 2 reveal the composition of the sol–gel liquid solution used to produce Ti_x_Si_y_ oxide. 

### 2.3. Characterization of Titanium–Silicon Oxide

#### 2.3.1. ^29^Si Solid-State Nuclear Magnetic Resonance Spectroscopy (^29^Si Solid-State NMR)

The ^29^Si solid-state NMR spectra were used to evaluate the structure of the silicate phase in SiO_2_ and Ti_x_Si_y_ oxide. ^29^Si NMR spectra were recorded on a Bruker Avance III HD 500 MHz spectrometer (Billerica, MA, USA).

#### 2.3.2. Fourier Transform Infrared Spectroscopy (FT-IR)

Infrared spectra of Ti_x_Si_y_ oxide, TiO_2_, and SiO_2_ were recorded by Fourier transform infrared spectroscopy (FTIR) (Bruker, Tensor 27, Billerica, MA, USA) using attenuated total reflectance (ATR) equipped with a platinum diamond crystal (TYPE A225/QL). The spectra were recorded at wavenumbers from 400 to 4000 cm^−1^ with 4 cm^−1^ resolution through the accumulation of 64 scans. All samples of Ti_x_Si_y_ oxide, TiO_2_, and SiO_2_ were dried in an oven at 70 °C for 4 h before testing.

#### 2.3.3. Field Emission Scanning Electron Microscopy (FE-SEM)

FE-SEM images of Ti_x_Si_y_ oxide, TiO_2_, and SiO_2_ were examined by a field-emission scanning electron microscope (FESEM-EDS, Carl Zeiss Auriga, Oberkochen, Germany). The film specimens were coated with carbon prior to the investigation. An acceleration voltage of 3 kV was used to collect SEM images of the samples.

#### 2.3.4. X-ray Diffraction (XRD)

The diffractograms of Ti_x_Si_y_ oxide, TiO_2_, and SiO_2_ were recorded via powder X-ray diffraction (Bruker, Model D2 phaser, Billerica, MA, USA), with CuKα radiation, scanning from 10° to 90° at a rate of 0.05°/s, with a current of 35 mV and 35 mA.

#### 2.3.5. X-ray Absorption Near-Edge Structure Spectroscopy (XANES) and Extended X-ray Absorption Fine Structure Spectroscopy (EXAFS)

All TiO_2_ and SiO_2_ standard compounds and Ti_x_Si_y_ oxide samples were analyzed by XANES and EXAFS spectroscopy at the Ti *K*-edge at beamline 8 of the electron storage ring (using an electron energy of 1.2 GeV, a bending magnet, beam current of 80–150 mA, and 1.1 to 1.7 × 10^11^ photons s^−1^) at the Synchrotron Light Research Institute (SLRI), Nakhon Ratchasima, Thailand [25]. Finely ground, homogenized powder of each sample was spread as a thin film (area 2.0 cm × 0.5 cm) and carefully dispersed with a spatula to yield a homogeneous particle distribution and to avoid hole effects on Kapton tape (Lanmar Inc., Northbrook, IL, USA) mounted on a sample holder. At least five spectra were obtained for each standard compound and oxide sample. All XANES and EXAFS spectra were measured in the transmission mode with ionization chamber detectors. For the acquisition of all spectra, a Ge (220) double crystal monochromator with an energy resolution (ΔE/E) of 2 × 10^−4^ was used. An energy range of 4936–5824 eV with an energy step of 2, 0.2, 0.05k eV was used for Ti *K*-edge spectra. The photon energy was calibrated against the *K*-edge of Ti foil at 4966 ± 0.2 eV. Finally, the normalized XANES and EXAFS data were processed and analyzed after background subtraction in the pre-edge and post-edge regions using a software package including (i) ATHENA (Chicago, IL, USA) for XAS data processing, (ii) ARTEMIS (Chicago, IL, USA) for EXAFS data analysis using theoretical standards from FEFF, and (iii) HEPHAESTUS (Chicago, IL, USA) software for a collection of beamline utilities based on tables of atomic absorption data. This package is based on the IFEFFIT library of numerical and XAS algorithms and is written in the Perl programming language using the Perl/Tk graphics toolkit [26].

#### 2.3.6. X-ray Fluorescence (XRF)

The chemical compositions of Ti_x_Si_y_ oxide, TiO_2_, and SiO_2_ were investigated by energy-dispersive XRF (EDS Oxford Instrument ED 2000, Abingdon, UK) with a Rh X-ray tube with a vacuum medium.

#### 2.3.7. Particle Size Distribution

The particle size distribution of Ti_x_Si_y_ oxide, TiO_2_, and SiO_2_ was evaluated by a diffraction particle size analyzer (DPSA) (Malvern Instruments, model Mastersizer 2000, Great Malvern, England). The Ti_x_Si_y_ oxide, TiO_2_, and SiO_2_ were dispersed in absolute ethanol and analyzed by a He-Ne laser. The average particle size distribution was investigated from the standard volume percentiles at 10, 50, and 90%. The average volume-weighted diameter was used to define the average particle size.

#### 2.3.8. Specific Surface Area (BET)

The specific surface areas of Ti_x_Si_y_ oxide, TiO_2_, and SiO_2_ were investigated by a nitrogen adsorption analyzer (BELSORP MINI II, model Bel-Japan, Osaka, Japan). The samples of of Ti_x_Si_y_ oxide, TiO_2_, and SiO_2_ were degassed at 300 °C for 3 h before measurement. The Brunauer, Emmett, Teller (BET) and Barrett, Joyner, Halenda (BJH) methods were used to calculate the specific surface area and pore size distribution, respectively.

#### 2.3.9. Photocatalytic Activity Characterization

PLA, PLA/TiO_2_, PLA/SiO_2_, and PLA/Ti_x_Si_y_ oxide composites with 3 wt.% of Ti_x_Si_y_ oxide were prepared by the solvent-casting method. The polymer was dissolved in solvent (10% *w/v*) by stirring. Firstly, Ti_x_Si_y_ oxide was dispersed in chloroform with ultrasonic treatment for 1 day. After this, PLA was added to the Ti_x_Si_y_ oxide and strictly stirred for 4 days. The dispersions of PLA composites were additionally ultrasonically treated for 1 h using a frequency of 42 kHz, four times per day. The treated dispersions were gently poured onto Petri dishes, and the solvent was evaporated at room temperature. The films were dried to a constant mass, dried at room temperature for 24 h, and kept in the oven at 40 °C for 4 h. Film thickness was measured by using a micrometer in eight replicates for each sample, from which an average value was attained. Pure PLA and nanocomposite films had a uniform thickness of 200–250 µm.

The photocatalytic activity of PLA composite films was evaluated in terms of the degradation of methylene blue (MB) under UV light according to Chinese standard GB/T 23762-2009. PLA composite films were placed in a flask and then 200 mL of MB solution (10 mg/L) was added. The flask was placed on a mechanical shaker at 50 rpm in a UV chamber with 4 UV lamps (LP Hg lamps, 8 watts, main light emission at 245 nm). The schematic design of the investigational setup for the photocatalytic process is presented in Figure 2. Then, 4 mL MB solution was collected every 60 min and analyzed using a UV–vis spectrophotometer (Cary300, Agilent Technology, CA, USA). In order to maintain the volume of MB solution in the flask, the samples were returned after each measurement. The maximum absorbance of MB occurred at 664 nm (Figure 3). The spectrometer was calibrated with a solution of MB at 1mg/L, 3mg/L, 5mg/L, 7mg/L, and 10mg/L concentrations, respectively. The calibration curve of methylene blue aqueous solutions is shown in Figure 4. In order to precisely predict the decomposition of MB under UV light, the PLA and composite films, as well as the solution, were stored in a black box without any photocatalyst. The concentration of MB was also collected every 60 min to estimate the absorption of MB.

## 3. Results

### 3.1. Effect of pH

^29^Si Solid-state NMR spectroscopy was used to investigate the structures of Ti_x_Si_y_ oxide (particularly for 1:1 atomic ratio) and SiO_2_ in order to determine the development of the framework of silicon atoms in these materials. The structural analysis of ^29^Si NMR is illustrated in Figure 5, which presents the spectra of the silica and the Ti_50_Si_50_ oxide. In general, the ^29^Si NMR spectra of oxides show three peaks for different environments for Si: isolated silanol groups (SiO)_3_Si–OH, Q^3^; geminal silanols (SiO)_2_Si–(OH)_2_, Q^2^; and silicon in the siloxane binding environment without hydroxyl groups (SiO)_4_Si, Q^4^ (Q^n^ represents a SiO_4_ unit, with n being the number of bridging in oxygen atoms) [27,28]. According to related references, the three resonance signals of pure SiO_2_ (synthesized by sol–gel method) appearing at −93.9, −103.3, and −110.2 ppm can be attributed to the configurations of Q^2^, Q^3^, and Q^4^, respectively. The spectrum of silica (Figure 5a) exhibits a chemical shift of −110 ppm, which is attributed to the Q^4^ units, and a minor peak appears at −103 ppm that would conform to the Q^3^ units corresponding to the siloxane (Si–O–Si) and silanol (≡Si–OH) groups of silica. The strength of Q^4^ structural units [Si(SiO)_4_] represents a three-dimensional network of silica, as evidenced by the intensity of the peaks in Figure 5a. There would also be a very small proportion of Si nuclei directly bonded to the hydroxyl groups, influenced by the chemical shift of the Q^3^ structural unit [Si(SiO)_3_OH] [17]. Additionally, Figure 5b indicates that silica was chemically combined with titanium oxide for the formation of the Ti_50_Si_50_ oxide, causing Ti atoms to be disrupted in the three-dimensional silica network and the microstructure surrounding the silicon atoms. When comparing the peak at approximately -103 ppm to the quantity of siloxane (Si–O–Si) and silanol (≡Si–OH) groups in pure SiO_2_, a very substantial increase in Q^3^ units (mono-substitution) is detected, which might be due to the existence of significant Si–O–Ti bonds in this sample, signaling atomic mixing. The presence of Ti–O–Si–O–Ti–O groups may have caused the appearance of Q^2^ units, which might be linked to disubstituted silica in tetrahedra [29]. A similar observation was reported by Pabón, Retuert, and Quijada (2007) [17]. Obviously, Ti atoms replaced some Si atoms in the SiO_2_ network, indicating that, as a consequence of the synthesis performed by the sol–gel method defined in this work, a large number of Si-O–Ti bonds were created, in agreement with the results of the FTIR and XRD. 

FTIR spectra of TiO_2_, SiO_2_, and Ti_50_Si_50_ oxides synthesized at various pH values are shown in Figure 6. TiO_2_ and all Ti_50_Si_50_ oxides displayed a prominent band at 400 to 700 cm^−1^ in their FTIR spectra, which corresponded to the bending and stretching mode of Ti–O–Ti and was characteristic of well-ordered TiO_6_ octahedrons [30]. 

Strong bands near 3440 and 1630 cm^−1^ can be seen in Figure 6, which are assigned to the streching and deformation vibration of the hydrocyl groups present (TiO_2_–OH) on the surfaces of TiO_2_ and all mixed oxide samples. The intensive band of OH-group asymmetrical and symmetrical stretching vibrations at 3440 cm^−1^ and O–H deformation vibration at 1630 cm^−1^ could confirm the large amount of water molecules [31]. However, the OH-group vibration bands were substantially weaker as the calcination temperatures increased [32]. The peaks at 440, 801, and 1052 cm^−1^ corresponded to the rocking, symmetric, and asymmetric stretching vibrations of Si–O–Si of silica, respectively [3]. However, TiO_2_ also presents a peak at 1000-1200 cm^−1^, which is assigned to the deformation vibration of Ti–O–Ti of TiO_2_ [33]. The peak at 949 cm^−1^ was commonly attributed to Ti–O–Si vibrations by various authors, indicating the existence of Ti–O–Si linkages [34]. This band can also be attributed to the Si–OH bond, which should be attributed to Si–OH bonding. However, this absorption band disappeared after the SiO_2_ particles were calcined at 450 °C. On the other hand, the 949 cm^−1^ absorption band was still observed for all Ti_x_Si_y_ oxide samples that had been calcined at a temperature of 450 °C. Accordingly, the 949 cm^−1^ peak in the FTIR spectra for Ti_x_Si_y_ oxide particles should not be attributed to Si–OH bonds but rather be associated with Si–O–Ti bonding [35,36]. This indicated titanium being combined into the framework of silica. Therefore, the influence of pH on the FTIR of the Ti_50_Si_50_ oxide was not observed. Based on the evidence from NMR and FTIR, the probable mechanism for the synthesized Ti_x_Si_y_ oxides based on results of earlier studies [37] is suggested in Figure 7.

Figure 8 shows FE-SEM images of SiO_2_ and Ti_50_Si_50_ oxide powder nanoparticles synthesized at various pH values. SiO_2_ particles were mostly spherical in shape, with low agglomeration (Figure 8a). Meanwhile, large bulk particles with high agglomeration of Ti_50_Si_50_ oxides are shown in Figure 8b. This agglomeration with an irregular shape was a result of the Ti and Si precursors in the sol solution during the processing, synthesized due to the absence of appropriate hydroxide ions (OH^−^ ions) at pH 8.0 [38]. Figure 8c,d show that the nanoparticles were spherical in shape and the size distribution of particles became more uniform, with a respectable nanostructure, upon increasing the pH to alkaline conditions (pH 9.0 and 10.0). In this work, Ti_x_Si_y_ oxide nanoparticles were essentially spherical in form, with low agglomeration. However, the size of the Ti_50_Si_50_ oxide particles at pH 10.0 was nano-sized, with the appearance of agglomeration (Figure 8d). Wu, Wu, and Lü, (2006) claimed that the colloidal sol–gel technique can generate large particles composed of agglomerated nanoparticles and either condensed or porous polycrystalline microparticles [39]. Therefore, pH 9.0 was selected to achieve the compositionally controlled synthesis of Ti_x_Si_y_ oxide particles that are spherical in shape.

XRD patterns of Ti_50_Si_50_ oxide powders synthesized at various pH values (8.0, 9.0, and 10.0) are shown in Figure 9. Silica showed one amorphous peak located at 2ϴ angles at 23°. In most of the Ti_50_Si_50_ oxide powders prepared at pH 8.0 and 9.0 (Figure 9), XRD patterns exhibited a broad amorphous peak because of the existence of silica. Furthermore, the formation of the Ti_50_Si_50_ oxides suggests that titanium oxide was combined with the silica by atomic mixing, and consequently the distribution of Ti atoms occurred in the microstructure around the silicon atoms and the three-dimensional silica network. However, the Ti_50_Si_50_ oxide powders prepared at pH 10.0 exhibited an intense TiO_2_ anatase peak, suggesting that the formation of higher crystallites of anatase occurred in Ti_50_Si_50_ oxides when the high pH of the sol–gel solution was applied during hydrolysis [38]. It can be clearly seen that the crystallinity of anatase in Ti_50_Si_50_ oxide samples is enhanced by adjusting the pH value of the sol. 

Furthermore, the XRD peaks for Ti_50_Si_50_ oxide powder with pH > 9 present a crystalline nature with 2ϴ angles at 25° (101) and 48° (200), which correspond to the crystalline phase (anatase) of TiO_2_ [3,34], due to the presence of a sufficient amount of OH^−^ to form TiO_2_. It is possible that the pH influences the particle size and crystallinity of Ti_50_Si_50_ oxides. The crystal structure of TiO_2_ was tetragonal; a = b = 3.782 Å, c = 9.502 Å [40]. However, no rutile phase was detected in any case, according to the absence of (110) diffraction peak at 27.4°.

Titanium K-edge X-ray-absorption near-edge structure (XANES) spectroscopy is widely used to derive information on the coordination environment of tetravalent Ti[Ti(IV)] in structurally complex oxide materials. Normalized Ti K-edge XANES spectra of anatase TiO_2_ standards, synthesized TiO_2_ materials, and Ti_50_Si_50_ oxides synthesized at various pH values were collected and are presented in Figure 10. The spectrum is mostly separated into two regions: (i) the pre-edge region (4940 to 4990 eV) and (ii) the post-edge region (4990 to 5010 eV) [41]. The anatase is composed of pre-edge features A_1_–A_3_ and a characteristic shoulder B [42,43]. The origin of these features was described by Farges et al. The first pre-edge (A_1_) is mainly attributed to quadrupolar transitions to t_2g_ levels of the TiO_6_ octahedron. The second pre-edge (A_2_) and the third pre-edge (A_3_) are attributed to 1s to 3d dipolar transitions [43]. The B features may be attributed to the interactions of the central Ti 4p orbital hybridized with the near Ti or O atom (Wu et al., 1997). The intensive absorbance of the B feature may be due the local structures of the Ti orbital hybridized for the TiO_2_–SiO_2_ photocatalyst [41]. Both TiO_2_ anatase and synthesized TiO_2_ display three minor pre-edge peaks, which are attributed to transitions from the 1s level of Ti to 1t_1g_, 2t_2g_, and 3e_g_ molecular orbitals (in sequence of increasing energy) [44]. The three minor pre-edge peaks for TiO_2_ anatase are outstanding fingerprints for the crystalline phase of titanium dioxide materials. Thus, the formation of anatase is specified by the near-edge region of TiO_2_.

As silicon merged into the Ti_50_Si_50_ oxide, a large single pre-edge feature was observed to dominate the other weaker feature. From Figure 10, the significant pre-edge feature in Ti_50_Si_50_ oxide samples indicates that a larger proportion of Ti atoms occupy tetrahedral symmetry positions, as expected for samples where Ti directly replaces Si in the oxide sample. Moreover, the pre-edge area of Ti_50_Si_50_ oxides was not affected by the various pH values applied during synthesis. Furthermore, the XANES spectra of the Ti_50_Si_50_ oxides have a higher intensity of pre-edge peaks than TiO_2_ anatase, indicating that Ti is no longer entirely confined to octahedral sites when combined with Si. Therefore, the results of the XANES spectra of mixed oxide samples are in reasonable agreement with the XRD results.

The EXAFS analysis at the Ti K-edge was used to investigate the coordination number of Ti atoms and bond distances between Ti–O and Ti–Si atoms. The program package of ATHENA–ARTEMIS was used to perform the background noise correction and the normalization of the raw data [26]. The EXAFS data were FT to the R-space to define the local atom structure and relative bond length with respect to the absorbing atom. The ATOM and FEFF codes were used to simulate the FT data (in R-space) by creating a systematic theoretical structure of SiO_2_ [45]. The range of data reserved for the transformation was 3–9 Å^−1^ in k-space. Structural parameters were attained, without the phase corrections, by fitting the data in R-space, within the intermission of 1.1–3 Å (Figure 11). The shell parameters, such as Ti–O and Ti–Si bond distances (R), coordination number (CN), Debye–Waller factor (σ^2^), and R-factor, are presented in Table 3. Note that the amplitude reduction factor ($S_{0}^{2}$) was preliminarily determined to be 0.52 by fitting the EXAFS spectrum of SiO_2_.

The first and second shells in the FTs arise from the single scattering paths of cation–oxygen and cation–cation, respectively [46]. In this study, we first focus on the Ti–O and Ti–Si shells of all Ti_x_Si_y_ oxide samples. TiO_2_ anatase has an average Ti–O bond distance of 1.95 Å. The results from curve fitting of the EXAFS provide an interatomic distance of 1.99, 1.82, and 1.94 Å for Ti atom dilute silica of Ti_50_Si_50_ oxides synthesized at various pH levels of 8.0, 9.0, and 10.0, respectively. This result can be compared favorably with the bond distance of Ti–O in the titanium silicalite molecular sieve of 1.80 Å, corresponding to the cation existing in a tetrahedral environment [47,48]. Neurock and Manzer (1996) calculated the Ti–O bond distance using density functional methods, obtaining a value of of 1.81 Å, suggesting Ti in a tetrahedral cluster [49]. On the other hand, we found that there is evidence of a longer Ti–O distance at 1.99 at pH 8.0, indicating a six-fold Ti site (octahedral), which agrees with Loshmanov et al. [47], who investigated TiO_2_ in SiO_2_ with neutron diffraction and attempted to isolate the Ti–O bond by subtracting their radial distribution functions from that of pure SiO_2_. They assigned a value of 1.95 A. By analogy with the Ti–O distance in its octahedrally bonded oxides (1.91–2.01 Å), they inferred an octahedral coordination for Ti in SiO_2_. This may be due to the effect of pH on the reaction of the Ti_x_Si_y_ oxides synthesized. Obviously, the value of 1.83 Å described for the Ti–O bond distance in Ti_x_Si_y_ oxides synthesized at pH 9.0 is agreeable with the absolute value of the Ti atoms occupied in a tetrahedral environment, which supports the comparable assumption from XANES. Furthermore, all of the Ti_50_Si_50_ oxides presented a small value for the Debye–Waller factor, corresponding to the structural disturbance in the materials [45]. The Debye–Waller factor is used to describe the influence of static and thermal disorder on the EXAFS spectrum. A large value for the σ^2^ can be caused by a variance in the ligand distances (static disorder), whereas small Debye–Waller factors artificially increase the ligand contribution [50,51].

The chemical compositions of TiO_2_, SiO_2_, and Ti_50_Si_50_ oxide synthesized at various pH values by XRF are shown in Table 4. The XRF analysis showed that Ti_50_Si_50_ oxide synthesized at pH 8.0 showed a higher percentage of Si (54.39%) than those synthesized at pH 9.0 (47.87%) and pH 10.0 (9.82%) because TTIP can be hydrolyzed more quickly than TEOS in a solution at a high pH. In contrast, the atomic content of Ti in Ti_50_Si_50_ oxide synthesized at various pH values increased with the increasing pH value of the sol (43.72%). 

The particle sizes of TiO_2_, SiO_2_, and Ti_50_Si_50_ oxides synthesized at various pH values as determined by the Zetasizer Nano ZS and SEM micrographs are shown in Table 5 and Figure 12. The mean particle size distributions of TiO_2_ and SiO_2_ were 40.3 and 147.8 nm, while Ti_50_Si_50_ oxide synthesized at pH 9.0 presented a mean particle size distribution of 136.2 nm. However, the mean particle size distribution of Ti_50_Si_50_ oxide synthesized at pH 8.0 and 10.0 covered a range of 350–650 nm due to the tendency towards agglomeration.

To determine the correct diameter of TiO_2_, SiO_2_, and Ti_x_Si_y_ oxide, SEM micrographs (Figure 8) were used for measurement with ImageJ. The number average diameter, d_n_, was calculated from a minimum of 200 particles according to the following equations [52]:(1)$$d_{n}=\frac{\sum_{i}n_{i}d_{i}}{\sum_{i}n_{i}}$$
(2)$$SD=\sqrt{\frac{\sum_{i}n_{i}{\left(d_{i}{-d}_{n}\right)}^{2}}{N}}$$
where d_i_ is the diameter of a particle and n_i_ is the total number of particles having diameter, d_i._

It can be observed that the number average diameter of the sample increased with the increasing pH value of the synthesis. 

The textural properties of the SiO_2_, TiO_2_, and Ti_50_Si_50_ oxides synthesized at various pH values are shown in Table 6. It can be observed that the surface areas (S_BET_) of the oxides were 177.02, 225.68, and 62.75 m^2^g^−1^ for Ti_50_Si_50_ synthesized at pH 8.0, 9.0, and 10.0, respectively. In addition, the mean pore diameter of the Ti_50_Si_50_ oxides was 2.56, 5.85, and 27.31 nm. Meanwhile, the S_BET_ of SiO_2_ and TiO_2_ was 116.90 and 74.07 m^2^g^−1^, with an average pore size of 37.08 and 12.75 nm.

A typical nitrogen adsorption–desorption isotherm of the SiO_2_, TiO_2_, and Ti_50_Si_50_ oxides synthesized at various pH values is presented in Figure 13. The isotherm of SiO_2_ is a type II curve according to the IUPAC isotherm classification, indicating that the obtained SiO_2_ particles contain macropores. An inflection point occurs near the completion of the first adsorbed monolayer, which is typical for non-porous or macroporous materials with a pore size of >50 nm. In addition, all Ti_50_Si_50_ oxides exhibited type II isotherms with pore diameters of 2.56, 5.85, and 27.31 nm. However, the isotherm of TiO_2_ is a type IV curve according to the IUPAC isotherm classification. An abrupt hysteretic loop is detected from this curve, which is typical for mesoporous materials with pores in the range of 2 to 50 nm that show capillary condensation and evaporation [53,54].

### 3.2. Effect of Ti/Si Ratio

FTIR spectra of TiO_2_, SiO_2_, and Ti_x_Si_y_ oxide synthesized at pH 9.0 with different Ti/Si ratios are shown in Figure 14. All Ti_x_Si_y_ oxides have three major peaks in their FTIR spectra, with wavenumbers of 801, 949, and 1052 cm^−1^. The peaks at 801 and 1052 cm^−1^ corresponded to symmetric and asymmetric stretching vibrations of Si-O-Si of silica, while the peak at 949 cm^−1^ was commonly attributed to Ti–O–Si vibrations, indicating the existence of Ti–O–Si linkages [34]. This suggested that titanium had been incorporated into the silica framework. Therefore, there was no effect of the Ti/Si atomic ratio on the FTIR of Ti_x_Si_y_ oxide.

The FE-SEM image of TiO_2_ exhibits a large majority of particles with high agglomeration, as shown in Figure 15a. Ti_70_Si_30_ and Ti_50_Si_50_ oxides show particles that are homogeneous with a good nanostructure and mostly spherical in shape when Ti/Si is more than 1 (Figure 15b,c). However, the size of Ti_40_Si_60_ oxide particles becomes gradually aggregated with increasing moles of Si (Figure 15d).

From the XRD patterns, only broad-range signals due to the uniform dispersion of titanium atoms onto the SiO_2_ network can be seen in the XRD patterns of Ti_70_Si_30_, Ti_50_Si_50_, and Ti_40_Si_60_ oxides (Figure 16). As a result, the absence of XRD diffraction signals suggests that anatase crystallization is still hindered by the titania aggregate’s segregation from the silica network [55].

Ti K-edge XANES spectra of standard TiO_2_ anatase, synthesized TiO_2_ materials, and Ti_x_Si_y_ oxides synthesized with different Ti/Si ratios were collected and are presented in Figure 17. Ti is expected to readily replace Si in the tetrahedral SiO_2_ framework at low Ti concentrations in Ti_40_Si_60_. The features and position in the rising edge of the B shoulders are clearer and the positions are lower. This suggests that the Ti–O–Ti linkages in Ti_x_Si_y_ oxides increase and are incomplete in anatase [21]. 

The structural parameters of Ti_x_Si_y_ oxide synthesized at pH 9.0 with varying Ti/Si ratios were attained without phase corrections by fitting the data in R-space within the intermission of 1.1–3 Å (Figure 18). The shell parameters, such as Ti–O and Ti–Si bond distances (*R*), coordination number (*CN*), Debye–Waller factor (σ^2^), and R-factor, are presented in Table 3. The amplitude reduction factor ($S_{0}^{2}$) was estimated to be 0.52 based on the EXAFS spectra of SiO_2_. According to the EXAFS curve fitting findings, the interatomic distances for Ti atom dilute silica of Ti_70_Si_30_, Ti_50_Si_50_, and Ti_40_Si_60_ oxide are 1.90, 1.83, and 1.82, respectively. The average Ti–O bond distance dropped from 1.90 to 1.82 when Ti in Ti_x_Si_y_ oxide increased. Furthermore, the 1.83 and 1.82 Ti–O bond lengths found for Ti_50_Si_50_ and Ti_40_Si_60_ oxide correspond with the absolute value of the Ti atoms occupied in a tetrahedral environment, validating the comparable assumption from XANES.

The proportion of Si atoms in Ti_70_Si_30_, Ti_50_Si_50_, and Ti_40_Si_60_ oxide was verified by XRF analysis to be 29.79, 47.87, and 60.19%, respectively. Ti content was 70.21, 52.13, and 39.81% in Ti_70_Si_30_, Ti_50_Si_50_, and Ti_40_Si_60_ oxide, respectively (Table 4).

Table 5 and Figure 19 display the particle sizes of TiO_2_, SiO_2_, and Ti_x_Si_y_ oxides synthesized at various Ti/Si ratios, as measured by the Zetasizer Nano ZS and SEM micrographs. The mean particle size distribution of Ti_70_Si_30_, Ti_50_Si_50_, and Ti_40_Si_60_ oxide was 573.1, 136.1, and 825.5 nm, respectively. In addition, SEM micrographs (Figure 15) were used to measure the correct diameter of TiO_2_, SiO_2_, and Ti_x_Si_y_ oxide by ImageJ. The number average diameter, d_n_, was estimated using Equations (1) and (2) from the minimum value given in [52]. Ti_70_Si_30_, Ti_50_Si_50_, and Ti_40_Si_60_ oxide had an average diameter of 149.3, 135.4, and 131.8 nm, respectively, as reported in Table 6. As the amount of Ti incorporated into the silica network increased, the number average diameter decreased.

The Ti_70_Si_30_, Ti_50_Si_50_, and Ti_40_Si_60_ oxides had surface areas (S_BET_) of 569.07, 225.68, and 68.34 m^2^g^−1^, respectively. The S_BET_ displayed an upward trend as the Ti content increased. According to a prior study, the S_BET_ of the Ti_x_Si_y_ oxide was enhanced as the Ti concentration in the silica network increased [56]. 

A typical nitrogen adsorption–desorption isotherm of the SiO_2_, TiO_2_, and Ti_70_Si_30_, Ti_50_Si_50_, and Ti_40_Si_60_ oxides is presented in Figure 20. Type II isotherms with pore diameters of 10.96, 5.85, and 21.97 nm were found in Ti_70_Si_30_, Ti_50_Si_50_, and Ti_40_Si_60_ oxides, corresponding to mesoporous materials with pore sizes ranging from 2 to 50 nm. However, the TiO_2_ isotherm is classified as a type IV curve according to the IUPAC. This curve shows an abrupt hysteretic loop, which is common in mesoporous materials with pores between 2 and 50 nm that exhibit capillary condensation and evaporation [53,54].

### 3.3. Photocatalytic Degradation of Methylene Blue (MB)

The degradation of methylene blue (MB) was used to study the photocatalytic activity of PLA and PLA/Ti_x_Si_y_ oxide composite films. In addition, UV irrigation may also lead to the decomposition of MB without the presence of any photocatalyst. The degradation of MB on PLA and PLA/Ti_x_Si_y_ oxide composite films was caused by a change in MB concentration in aqueous solution under UV irrigation, as illustrated in Figure 21 and Figure 22. The addition of 3%wt TiO_2_, SiO_2_, and Ti_x_Si_y_ oxide to the PLA film matrix increased the effectiveness of photocatalysis in the degradation of MB. TiO_2_ is more effective at degrading MB than Ti_50_Si_50_ pH 9.0, Ti_50_Si_50_ pH 10.0, Ti_50_Si_50_ pH 8.0, and SiO_2_. In the degradation of MB, the efficiency of the materials was of the following order: TiO_2_ > Ti_70_Si_30_ > Ti_50_Si_50_ > Ti_40_Si_60_ > SiO_2_.

Photocatalytic activity is generally accepted to occur at the surface of a photocatalyst. As a result, the surface area of PLA nanocomposite film, which is dependent on the size of nanoparticles, film morphology, and thickness, influences photocatalytic reactivity [7]. The PLA nanocomposite film including TiO_2_ degrades MB more efficiently than employing only photocatalysis because of the two mechanisms of degradation. The first is that in which UVC directly degrades MB, while the second occurs when TiO_2_ receives light energy greater than the bandgap energy, and electrons in the valence band (VB) are excited into the conduction band (CB), resulting in the generation of a hole (h^+^) (Equation (3)). This hole can oxidize MB (Equation (4)) or oxidize H_2_O to produce OH (Equation (5)). The e^−^ in CB can reduce O_2_ at the surface of TiO_2_ to generate $O_{2}^{-}$ (Equation (6)). Radicals (OH, $O_{2}^{-}$) and h^+^ interact with MB to create peroxide derivatives and hydroxylate or completely degrade to CO_2_ and H_2_O [7]. The photodegradation pathway can be summarized by Equations (3)–(6).
(3)$$TiO_{2}+UVC\;\to \;e^{-}+h^{+}$$
(4)$$h^{+}+MB\;\to \;CO_{2}+H_{2}O$$
(5)$$h^{+}+H_{2}O\;\to \;H^{+}+OH$$
(6)$$e^{-}+O_{2}\;\to \;O_{2}^{-}$$

## 4. Conclusions

Titania–silica oxides (Ti_x_Si_y_ oxides) can be prepared by the sol–gel technique. The effect of pH and the Ti/Si atomic ratio of titanium–silicon binary oxide (Ti_x_Si_y_) on the structural characteristics of Ti_x_Si_y_ oxide were investigated by using the sol–gel technique. ^29^Si solid-state NMR and FTIR measurements indicated that certain Si atoms in the SiO_2_ network had been replaced by Ti atoms, implying the formation of Si–O–Ti connections. As the pH was elevated to alkaline conditions (pH 9.0 and 10.0), the nanoparticles of Ti_50_Si_50_ oxide acquired a more spherical shape and their size distribution became more uniform, resulting in an acceptable nanostructure. Agglomeration was reduced in Ti_50_Si_50_ oxide nanoparticles, which were mostly spherical in form. However, the Ti_50_Si_50_ oxide particles at pH 10.0 become nano-sized and agglomerated. In Ti_50_Si_50_ oxide samples, the existence of a large pre-edge feature indicated that a higher percentage of Ti atoms occupied tetrahedral symmetry positions, as expected in samples where Ti directly substitutes Si. The result of 1.83 Å for the Ti–O bond distance in Ti_x_Si_y_ oxides generated at pH 9.0 accords with the fraction of Ti atoms occupied in a tetrahedral environment. The average diameter of the sample increased with the increasing pH of the sol. In addition, Ti_70_Si_30_, Ti_50_Si_50_, and Ti_40_Si_60_ oxide had an average diameter of 149.3, 135.4, and 131.8 nm, respectively. As the amount of Ti incorporated into the silica network increased, the average diameter decreased. The S_BET_ displayed an upward trend as the Ti content increased. Type II isotherms with pore diameters of 10.96, 5.85, and 21.97 nm were found in Ti_70_Si_30_, Ti_50_Si_50_, and Ti_40_Si_60_ oxides, corresponding to mesoporous materials with pore sizes ranging from 2 to 50 nm, which is common in mesoporous materials. However, the TiO_2_ isotherm was classified as a type IV curve. The addition of 3%wt TiO_2_, SiO_2_, and Ti_x_Si_y_ oxide to the PLA film matrix increased the effectiveness of photocatalysis in the degradation of MB. TiO_2_ was more effective at degrading MB than Ti_50_Si_50_ pH 9.0, Ti_50_Si_50_ pH 10.0, Ti_50_Si_50_ pH 8.0, and SiO_2_. In the degradation of MB, the order of efficiency is TiO_2_ > Ti_70_Si_30_ > Ti_50_Si_50_ > Ti_40_Si_60_ > SiO_2_. Moreover, PLA/Ti_70_Si_30_ also increased the effectiveness of the photocatalytic activity of PLA. Finally, PLA/Ti_70_Si_30_ improved the efficiency of the photocatalytic activity of PLA.

## Figures and Tables

**Figure 1 polymers-14-02729-f001:**
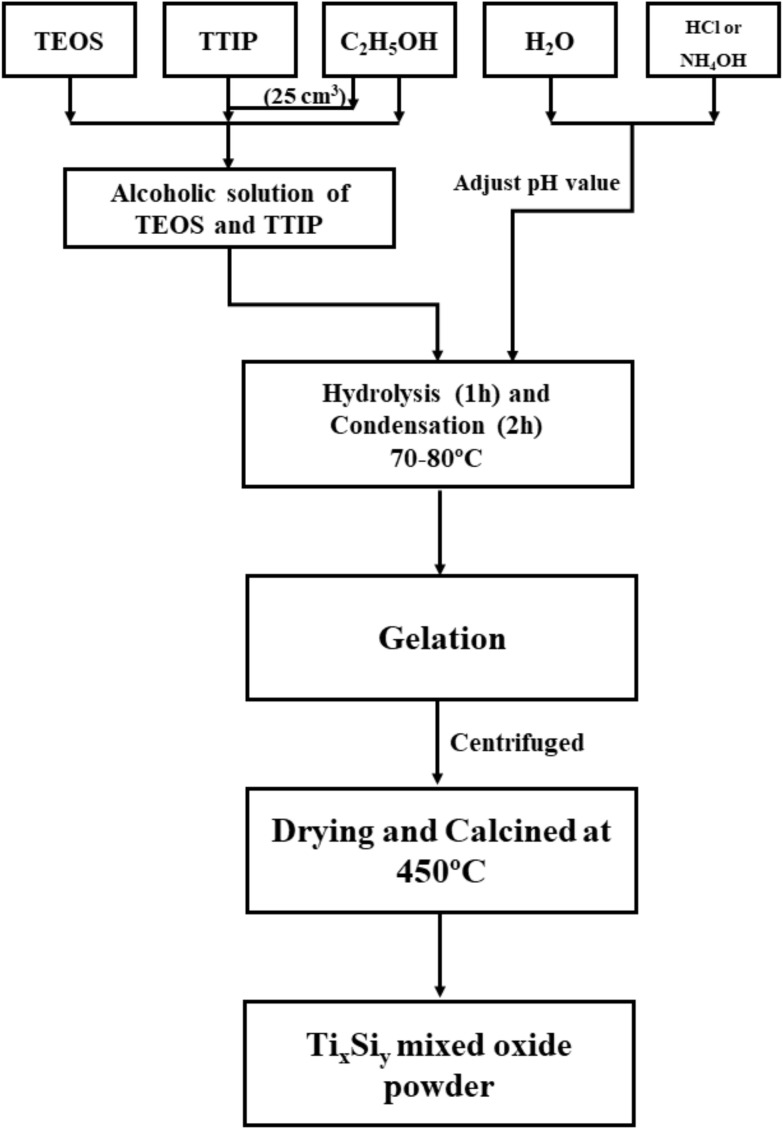
Experimental procedures for preparing Ti_x_Si_y_ oxide particles.

**Figure 2 polymers-14-02729-f002:**
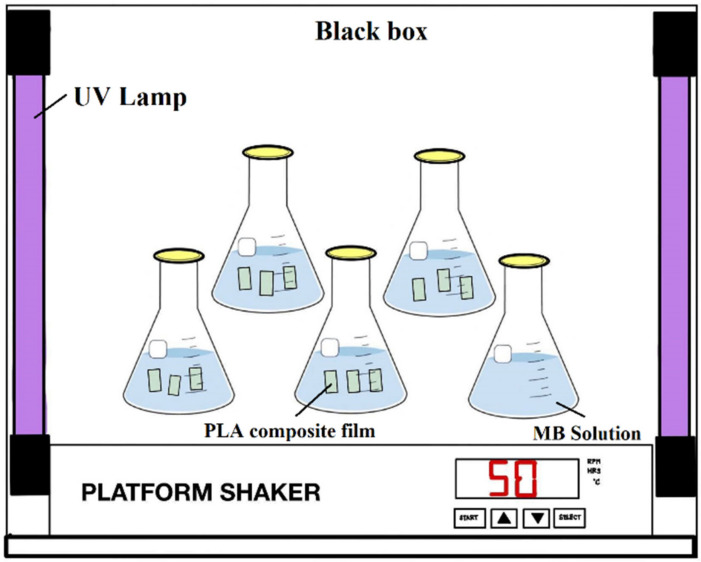
Schematic illustration of the experimental setup for photocatalytic process.

**Figure 3 polymers-14-02729-f003:**
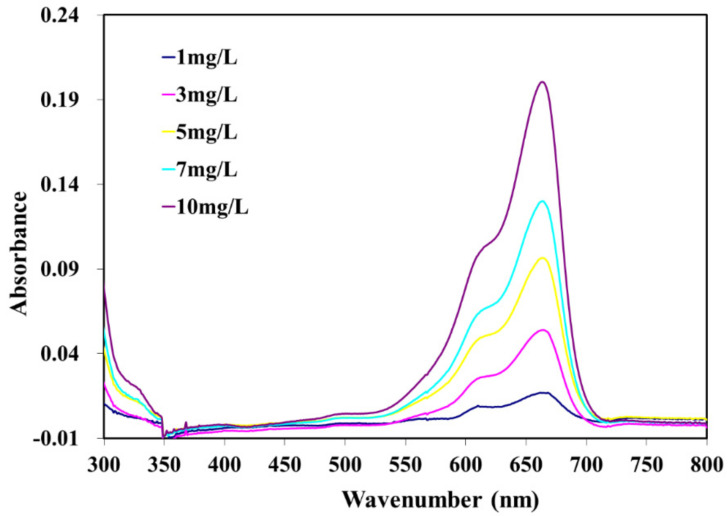
Wavelength–absorbance curves of methylene blue (MB) aqueous solutions.

**Figure 4 polymers-14-02729-f004:**
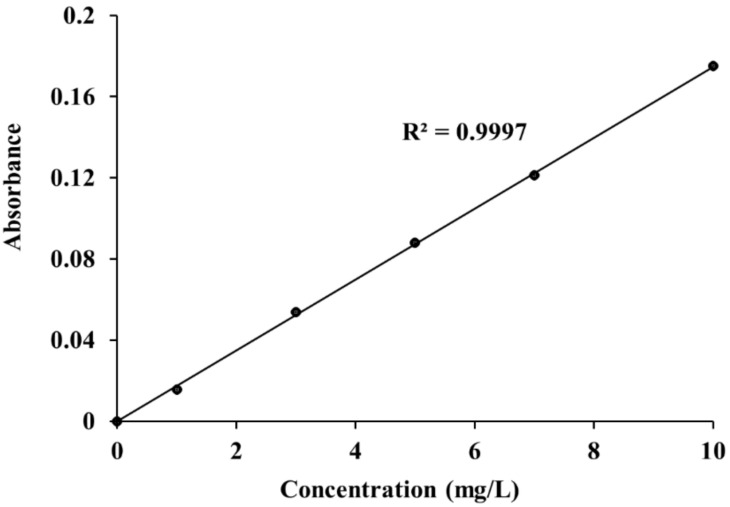
Calibration curve of methylene blue (MB) aqueous solutions.

**Figure 5 polymers-14-02729-f005:**
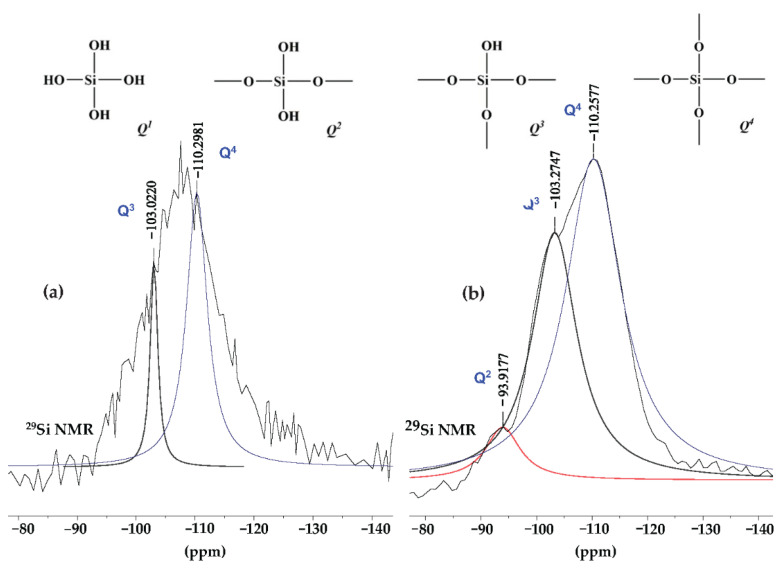
^29^Si-NMR patterns of (**a**) SiO_2_ and (**b**) Ti_50_Si_50_ oxide.

**Figure 6 polymers-14-02729-f006:**
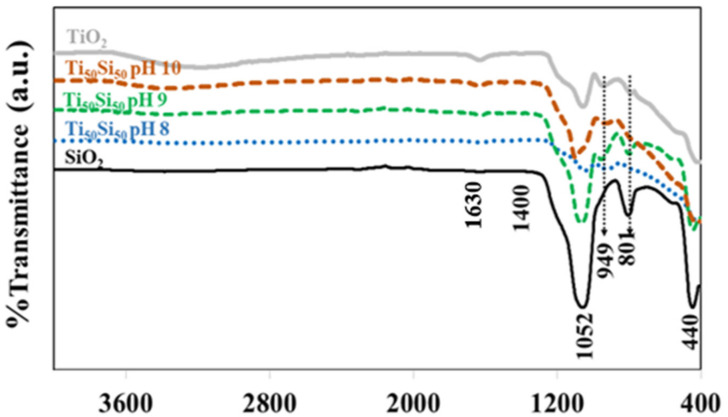
FTIR spectra of TiO_2_, SiO_2_, and Ti_x_Si_y_ oxides synthesized at pH 8.0, 9.0, and 10.0.

**Figure 7 polymers-14-02729-f007:**
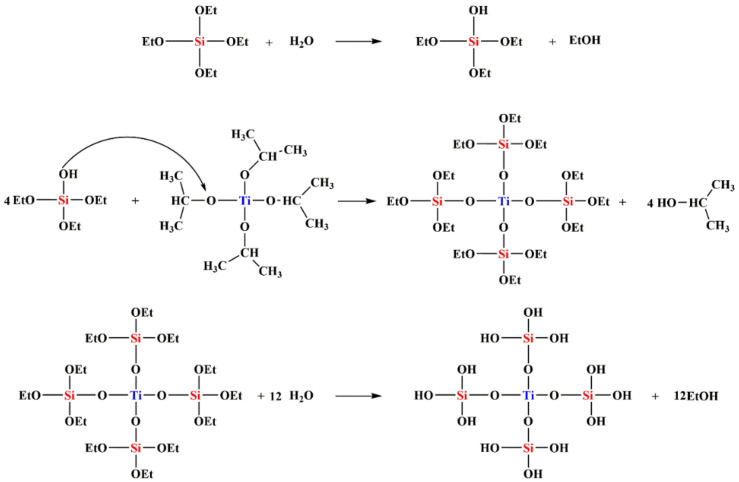
Proposed probable mechanism for synthesized Ti_x_Si_y_ oxides.

**Figure 8 polymers-14-02729-f008:**
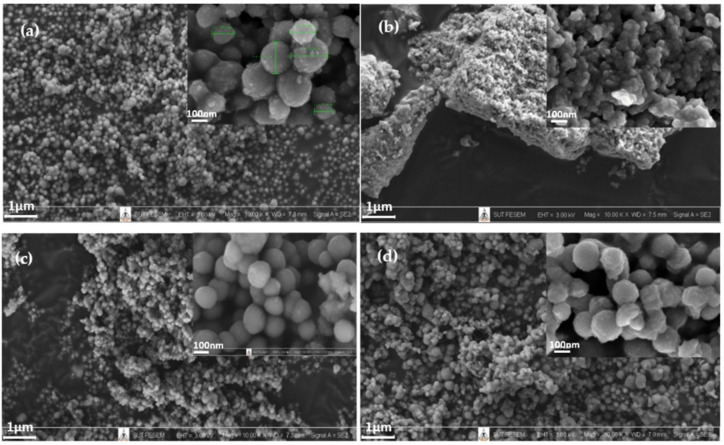
Low (×10k WD = 7.0–7.5 mm EHT = 3.00 kV) and high (×100k WD = 6.9–7.3 mm EHT = 3.00 kV) magnification SEM images of Ti_50_Si_50_ oxides synthesized at various pH values: (**a**) SiO_2_, (**b**) Ti_50_Si_50_ pH 8.0, (**c**) Ti_50_Si_50_ pH 9.0, (**d**) Ti_50_Si_50_ pH 10.0.

**Figure 9 polymers-14-02729-f009:**
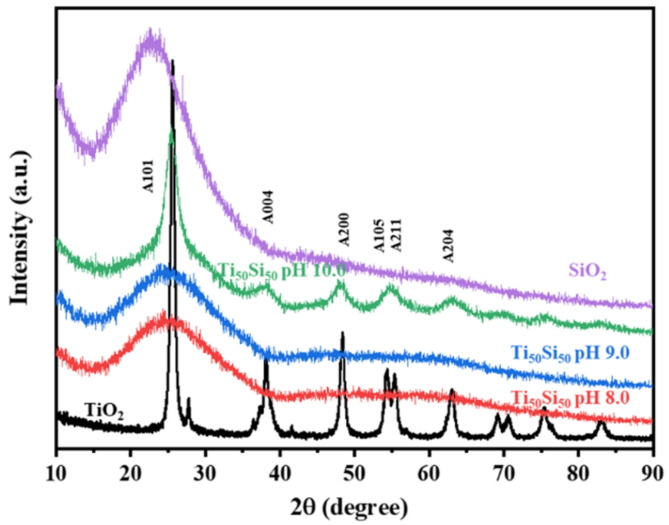
XRD patterns of Ti_50_Si_50_ oxide synthesized with pH 8.0, 9.0, and 10.0.

**Figure 10 polymers-14-02729-f010:**
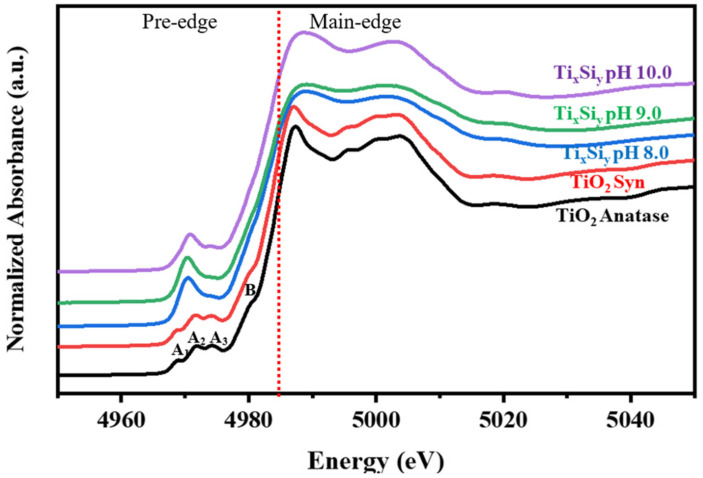
Ti *K*-edge XANES spectra of Ti_50_Si_50_ oxide synthesized with pH 8.0, 9.0, and 10.0.

**Figure 11 polymers-14-02729-f011:**
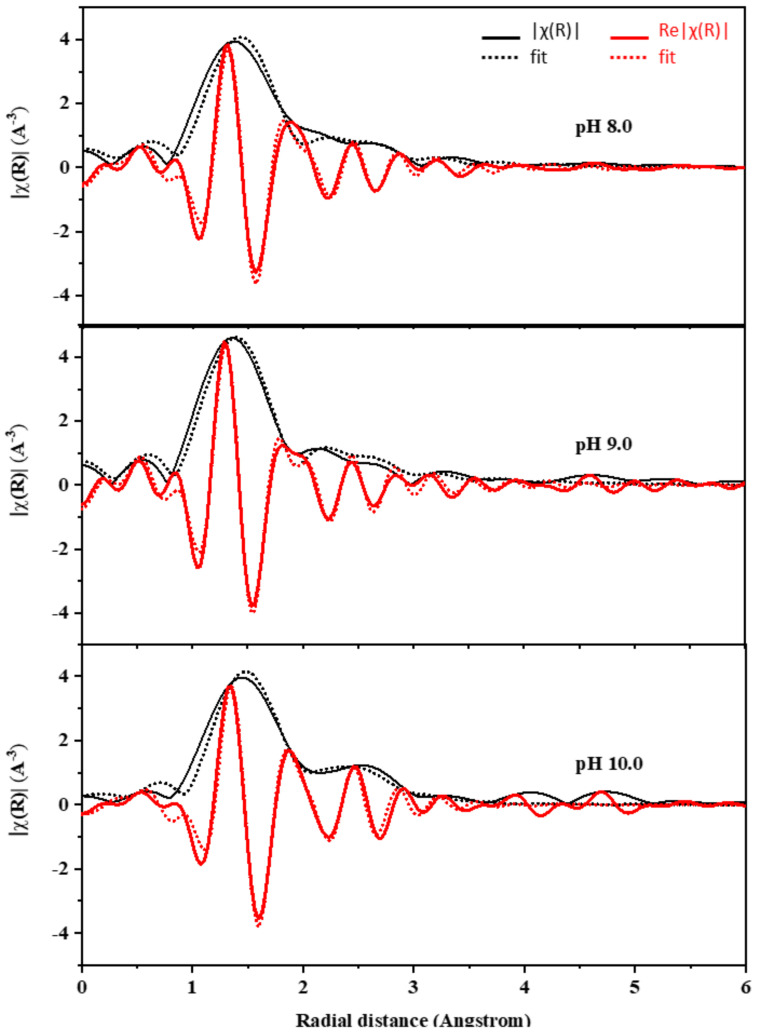
Ti *K*-edge XANES spectra of Ti_50_Si_50_ oxide synthesized at pH 8.0, 9.0, and 10.0.

**Figure 12 polymers-14-02729-f012:**
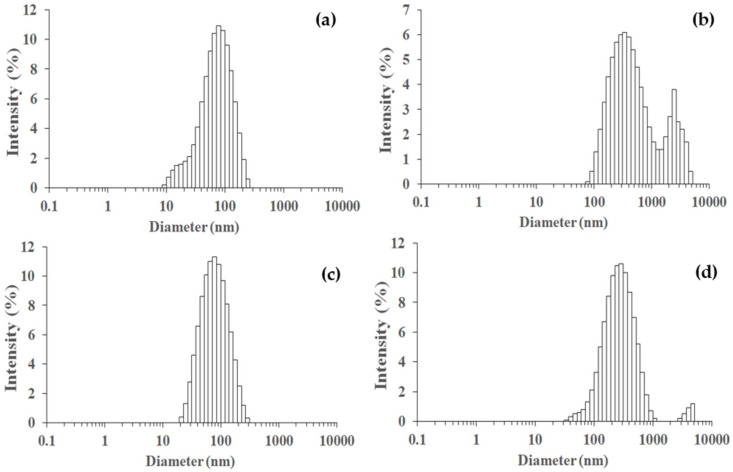
Particle size distributions measured on a Zetasizer Nano ZS of SiO_2_ and Ti_50_Si_50_ oxide synthesized at various pH values: (**a**) SiO_2_, (**b**) Ti_50_Si_50_ pH 8.0, (**c**) Ti_50_Si_50_ pH 9.0, (**d**) Ti_50_Si_50_ pH 10.0.

**Figure 13 polymers-14-02729-f013:**
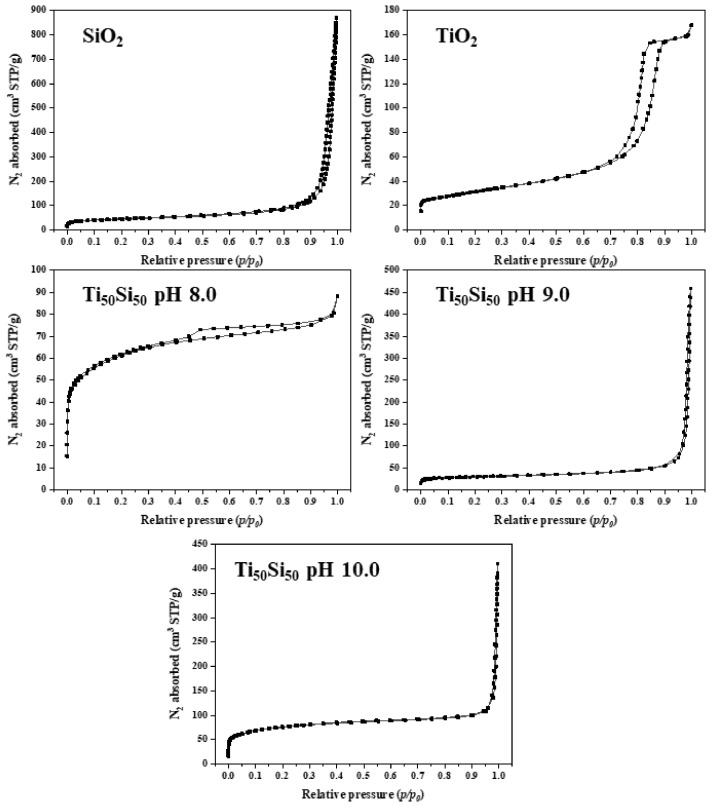
Nitrogen adsorption–desorption isotherms of SiO_2_, TiO_2_, and Ti_50_Si_50_ oxide synthesized at various pH values.

**Figure 14 polymers-14-02729-f014:**
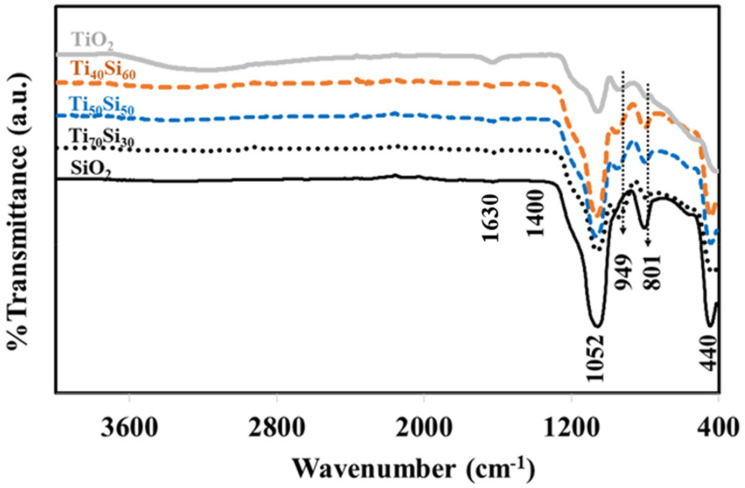
FTIR spectra of TiO_2_, SiO_2_, and Ti_x_Si_y_ oxide synthesized at pH 9.0 with different Ti/Si ratios.

**Figure 15 polymers-14-02729-f015:**
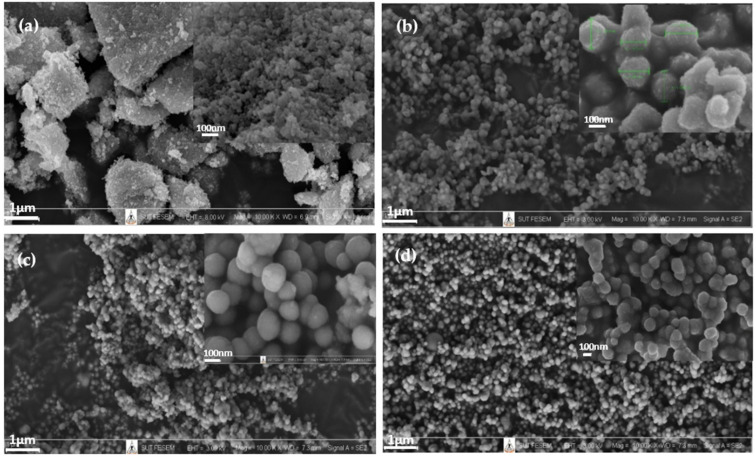
Low (×10k WD = 7.3 mm EHT = 3.00 kV) and high (×100k WD = 6.9–7.3 mm EHT = 3.00 kV) magnification SEM images of TiO_2_ and Ti_x_Si_y_ oxide synthesized at pH 9.0 with different Ti/Si ratios: (**a**) TiO_2_, (**b**) Ti_70_Si_30_, (**c**) Ti_50_Si_50_, (**d**) Ti_40_Si_60_.

**Figure 16 polymers-14-02729-f016:**
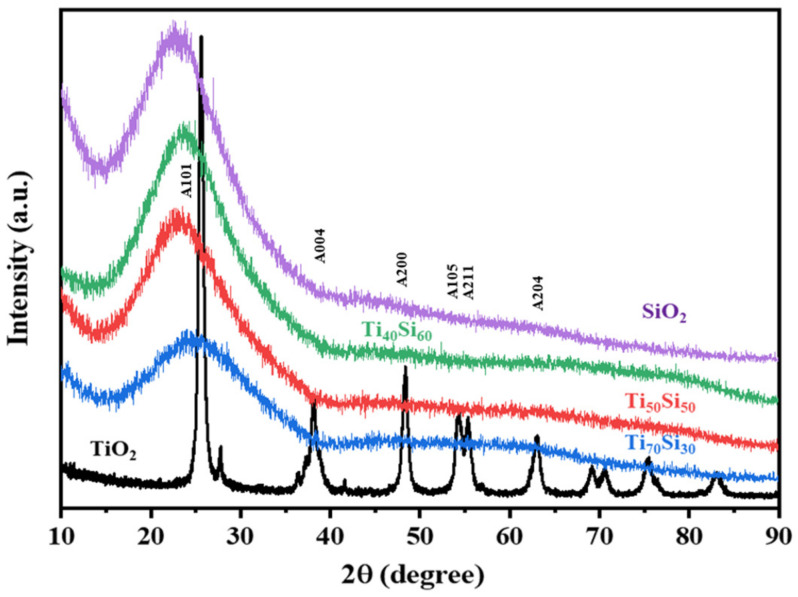
XRD patterns of SiO_2_, TiO_2_, and Ti_x_Si_y_ oxide synthesized at pH 9.0 with different Ti/Si ratios.

**Figure 17 polymers-14-02729-f017:**
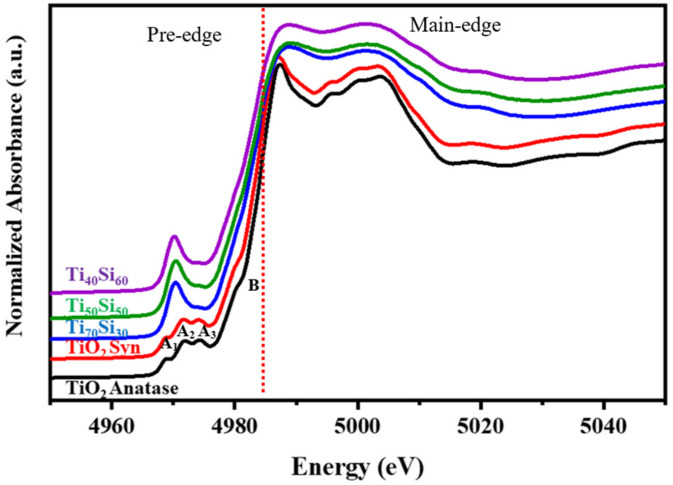
Ti *K*-edge XANES spectra of TiO_2_ and Ti_x_Si_y_ oxide synthesized at pH 9.0 with different Ti/Si ratios.

**Figure 18 polymers-14-02729-f018:**
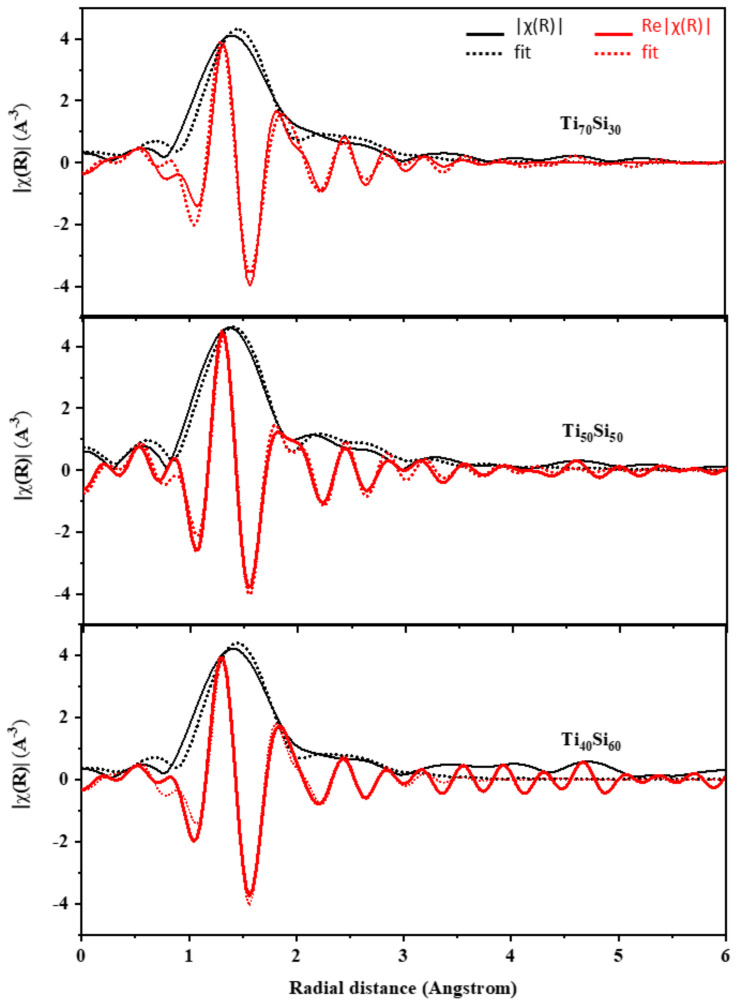
The experimental EXAFS data of Ti_x_Si_y_ oxide synthesized at pH 9.0 with different Ti/Si ratios in R-space within the intermission of 1.1–3 Å.

**Figure 19 polymers-14-02729-f019:**
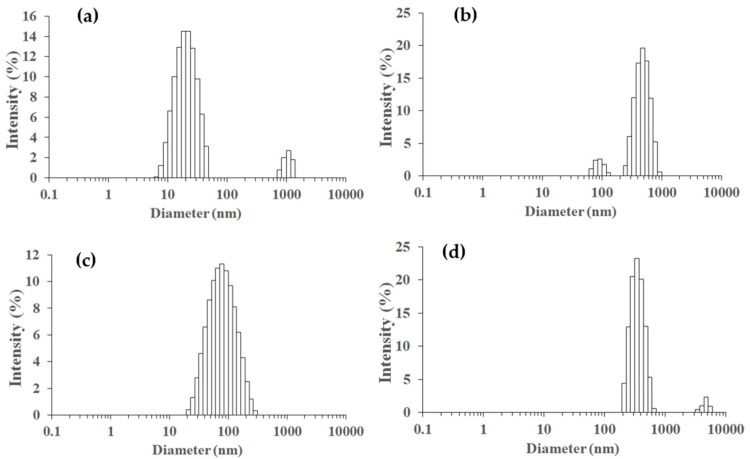
Particle size distributions measured on a Zetasizer Nano ZS of TiO_2_ and Ti_x_Si_y_ oxide synthesized at pH 9.0 with different Ti/Si ratios: (**a**) TiO_2_, (**b**) Ti_70_Si_30_, (**c**) Ti_50_Si_50_, (**d**) Ti_40_Si_60_.

**Figure 20 polymers-14-02729-f020:**
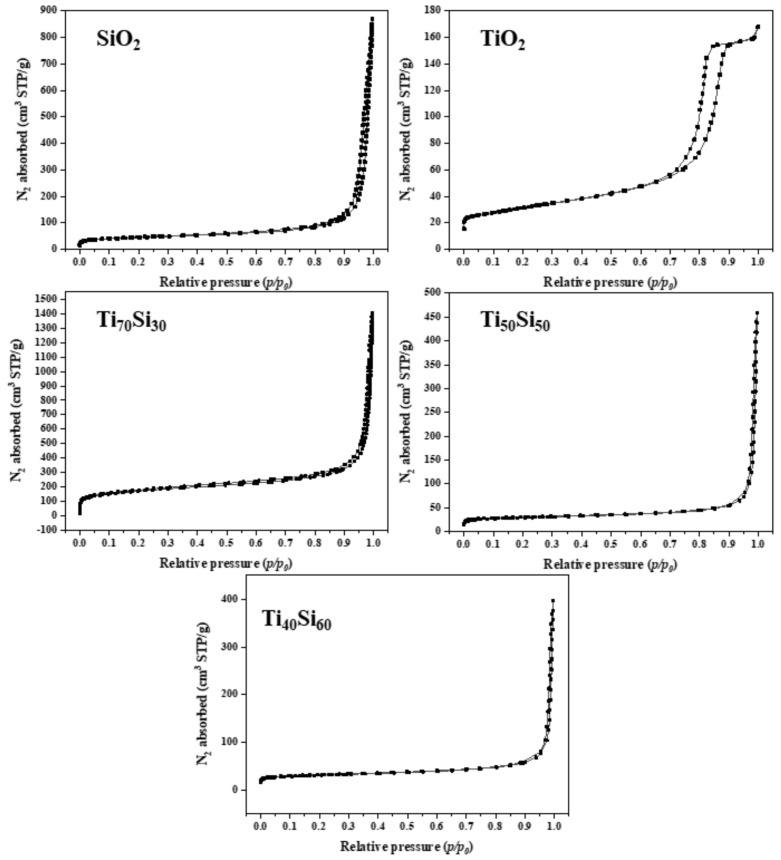
Nitrogen adsorption–desorption isotherms of SiO_2_, TiO_2_, and Ti_x_Si_y_ oxide synthesized at pH 9.0 with different Ti/Si ratios.

**Figure 21 polymers-14-02729-f021:**
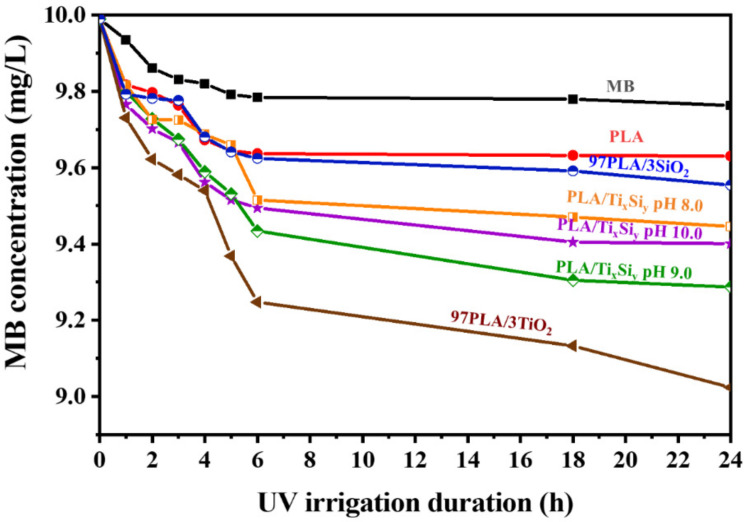
Concentration of methylene blue (MB) due to absorption of PLA film and PLA/Ti_50_Si_50_ oxide (synthesized at pH 8.0, 9.0, and 10.0) films under UV irrigation.

**Figure 22 polymers-14-02729-f022:**
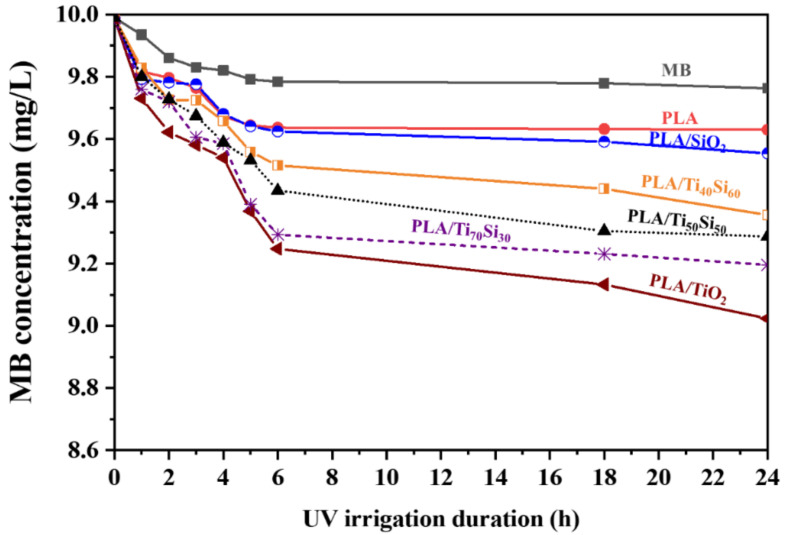
Concentration of methylene blue (MB) due to absorption of PLA film and PLA/Ti_x_Si_y_ oxide (synthesized at pH 9.0 with different Ti/Si ratios) films under UV irrigation.

**Table 1 polymers-14-02729-t001:** Moles of components in the preparation of Ti_50_Si_50_ oxide synthesized at various pH values.

Samples	TTIP (mol)	TEOS (mol)	C_2_H_5_OH (mol)	HCl/NH_4_OH (mol)	H_2_O (mol)
SiO_2_	-	0.120	1.889	0.060 ^b^	0.440
Ti_50_Si_50_ pH 8.0	0.011	0.109	1.889	0.025 ^b^	0.440
Ti_50_Si_50_pH 9.0	0.011	0.109	1.889	0.060 ^b^	0.440
Ti_50_Si_50_ pH 10.0	0.011	0.109	1.889	0.075 ^b^	0.440
TiO_2_	0.120	-	4.293	0.081 ^a^	0.702

^a^ moles of HCl, ^b^ moles of NH_4_OH.

**Table 2 polymers-14-02729-t002:** Moles of components in Ti_x_Si_y_ oxide synthesized at pH 9.0 with different Ti/Si ratios.

Samples	TTIP (mol)	TEOS (mol)	C_2_H_5_OH (mol)	NH_4_OH/HCl (mol)	H_2_O (mol)
SiO_2_	-	0.120	1.889	0.060 ^b^	0.440
Ti_70_Si_30_	0.020	0.100	1.889	0.060 ^b^	0.440
Ti_50_Si_50_	0.011	0.109	1.889	0.060 ^b^	0.440
Ti_40_Si_60_	0.008	0.112	1.889	0.060 ^b^	0.440
TiO_2_	0.120	-	4.293	0.081 ^a^	0.702

^a^ moles of HCl, ^b^ moles of NH_4_OH.

**Table 3 polymers-14-02729-t003:** Structural parameters (obtained from fittings of Ti *K*-edge EXAFS data) of first two coordination shells around Ti atom of Ti and Si in Ti_50_Si_50_ oxide synthesized at various pH values and Ti_x_Si_y_ oxide synthesized at pH 9.0 with different Ti/Si ratios.

Sample	Shell	Bond	ΔE (eV)	CN	R(Å)	σ^2^	R-Factor
**Ti_50_Si_50_** **pH 8.0**	1	Ti-O	4.11 ± 3.34	4.14 ± 0.18	1.99 ± 0.02	0.0053 ± 0.0013	0.0176
	2	Ti-Si		1.27 ± 0.44	2.79 ± 0.03	0.0018 ± 0.0019	
**Ti_50_Si_50_** **pH 9.0**	1	Ti-O	2.72 ± 2.03	4.07 ± 0.12	1.83 ± 0.01	0.0042 ± 0.0008	0.0082
	2	Ti-Si		1.00 ± 0.13	2.77 ± 0.02	0.0020 ± 0.0020	
**Ti_50_Si_50_** **pH 10.0**	1	Ti-O	6.70 ± 3.29	4.08 ± 0.19	1.94 ± 0.02	0.0049 ± 0.0024	0.0158
	2	Ti-Si		1.52 ± 0.22	2.81 ± 0.03	0.0013 ± 0.0031	
**Ti_70_Si_30_**	1	Ti-O	6.68 ± 3.23	4.15 ± 0.19	1.90 ± 0.02	0.0051 ± 0.0014	0.0198
	2	Ti-Si		0.97 ± 0.21	2.800 ± 0.04	0.0006 ± 0.0050	
**Ti_50_Si_50_**	1	Ti-O	2.72 ± 2.03	4.07 ± 0.12	1.83 ± 0.01	0.0042 ± 0.0008	0.0082
	2	Ti-Si		1.00 ± 0.13	2.77 ± 0.02	0.0020 ± 0.0020	
**Ti_40_Si_60_**	1	Ti-O	6.54 ± 2.27	4.09 ± 0.13	1.82 ± 0.01	0.0048 ± 0.0010	0.0103
	2	Ti-Si		1.19 ± 0.03	2.79 ± 0.03	0.0023 ± 0.0017	

**Table 4 polymers-14-02729-t004:** Percentage of atomic and atomic ratio of Ti and Si in TiO_2_, SiO_2_, and Ti_50_Si_50_ oxide synthesized at various pH values and Ti_x_Si_y_ oxide synthesized at pH 9.0 with different Ti/Si ratios.

Sample	Atomic (%)		Atomic Ratio of Ti/Si
	Si	Ti	
**SiO_2_**	100	0	-
**Ti_50_Si_50_** **pH 8.0**	54.39	45.61	0.84
**Ti_50_Si_50_ pH 9.0**	47.87	52.13	1.09
**Ti_50_Si_50_ pH 10.0**	9.82	93.18	9.49
**Ti_70_Si_30_**	29.79	70.21	2.36
**Ti_50_Si_50_**	47.87	52.13	1.09
**Ti_40_Si_60_**	60.19	39.81	0.66
**TiO_2_**	0	100	-

**Table 5 polymers-14-02729-t005:** Particle sizes of SiO_2_, TiO_2_, and Ti_50_Si_50_ oxide synthesized at various pH values, and Ti_x_Si_y_ oxide synthesized at pH 9.0 with different Ti/Si ratios.

Sample	Zetasizer Nano ZS		SEM
	Particle Size, d (nm)	Polydispesity Index, PDI	Value of Average Diameter, d_n_ (nm)
**SiO_2_**	147.8	0.248	144.2 ± 11.3
**Ti_50_Si_50_** **pH 8.0**	639.3	0.627	41.5 ± 16.7
**Ti_50_Si_50_** **pH 9.0**	136.2	0.239	135.4 ± 12.3
**Ti_50_Si_50_** **pH 10.0**	397.0	0.322	114.2 ± 24.2
**Ti_70_Si_30_**	573.1	0.630	149.3 ± 15.5
**Ti_50_Si_50_**	136.2	0.239	135.4 ± 12.3
**Ti_40_Si_60_**	825.5	0.907	131.8 ± 13.3
**TiO_2_**	40.3	0.490	27.8 ± 6.3

**Table 6 polymers-14-02729-t006:** Specific surface area, pore volume, and mean pore size of TiO_2_, SiO_2_, and Ti_50_Si_50_ oxide synthesized at various pH values, and Ti_x_Si_y_ oxide synthesized at pH 9.0 with different Ti/Si ratios.

Sample	Specific Surface Area, S_BET_ (m^2^g^−1^)	Pore Volume, V_p_ (cm^3^g^−1^)	Mean Pore Diameter, d_p_ (nm)
**SiO_2_**	116.90	1.06	51.08
**Ti_x_Si_y_ pH 8.0**	177.02	0.11	2.56
**Ti_x_Si_y_ pH 9.0**	225.68	0.33	5.85
**Ti_x_Si_y_ pH 10.0**	62.75	0.42	27.31
**Ti_70_Si_30_**	569.07	1.42	10.96
**Ti_50_Si_50_**	225.68	0.33	5.85
**Ti_40_Si_60_**	68.34	0.36	21.97
**TiO_2_**	74.07	0.23	12.75

## Data Availability

Not applicable.

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
