# Peer review of "Structural Characterization of Titanium–Silica Oxide Using Synchrotron Radiation X-ray Absorption Spectroscopy"

_polymers, 2022, doi:10.3390/polym14132729_

Round 1

Reviewer 1 Report

The paper is interesting and well written; however, there are some issues, particularly regarding the interpretation of the FTIR spectra.

A list of the main corrections request is listed below:

-          Lines 15-17 “The effect of pH 15 and the Ti/Si atomic ratio of titanium-silicon binary oxide (TixSiy) on the structural characteristics of TixSiy oxide are investigated by using the sol-gel technique”

Actually the sol-gel technique is used to prepare the samples, not to investigate their properties ; the sentence should be rephrased.

-          The TTIP/TEOS ratios for sample Ti50Si50 prepared at pH=9, presented Table 1 and 2, should be the same but they are not. The authors should explain.

-          In Figure 6 and 14, the FTIR spectra of TiO2 and SIO2 look very much alike. In the discussion of the FTIR spectra of the Ti50Si50 samples (lines 307-322), the band at 1052 cm-1 is attributed to asymmetric stretching vibrations of Si-O-Si of silica, but this band is also present in the spectrum of TiO2. On the other hand, the authors mention a Ti-O-Ti band that should be located at 1400 cm-1, but this band is absent in the spectrum of TiO2. I think this part of the paper should be checked and rewritten.

In particular, the sentence “the absence of the Ti-O-Ti bond in the oxide samples may be too weak to be detected at 1400 cm-1” (lines 319-320) should be rewritten.

-          In the SEM images (figure 8 and 15), the enlargement used are shown in the picture, but they are almost impossible to read; the authors could indicate the enlargement used in the figure caption also.

-          In the discussion of the XRD results (lines 373-384) the XRD patterns of sample Ti50Si50pH10 reveal the formation of anatase. I have one question to the authors: shouldn’t anatase formation be detectable from FTIR spectra too?

-          In the discussion of the EXAFS result (lines 450-465 and table 3), the Ti-O bond distance for sample Ti50Si50pH8 (1.99 Å) does not seem to agree with the value expected for ti-O bond distance in a tetrahedral environment (1.83 Å) and closer to the value expected for anatase (1.95 Å); the authors should comment on that.

-          Concerning the determination of the particle diameter (lines 504-513), I guess SD (line 509) stands for standard deviation and N=Sini; what does PDI in table 5 stand for?

-          Line 628-630 “However, the pre-edge peaks for TixSiy oxides are not as powerful as those predicted for Ti in a completely tetrahedral environment.” A citation is needed.

Other minor corrections are listed below:

Line 15 “Titania-silica oxides (TixSiy oxides) was prepared” change to “Titania-silica oxides (TixSiy oxides) were prepared”

Line 25 “ resulting in a respectable nanostructure.” Respectable is a strange adjective; I suggest using another one.

Line 27 “a significant pre-edge feature in Ti50Si50 oxide samples” ; change to “a significant pre-edge feature in the spectra of Ti50Si50 oxide samples” 

Line 33”degrading MB.” Add: (methylene blue)

Line 34 “PLA/ i70Si30” change to:  “PLA/ Ti70Si30

Line 48 “Titanium dioxide” change to: “titanium dioxide”

Line 78 “factors to determining” change to: “factors determining”

Line 210 “200 mL of MB solution (10 mg/L) was added” change to:  “200 mL of MB solution (10 mg/L) were  added”

Line 291 “was reported by Pabón, Retuert, and Quijada (2007).” I guess it is reference [17] and so it should b cited in the text 8if it is not, it should be added to the References list”

 Line 294 “bonds was created” change to “bonds were created”

Line 368 “Wu, Wu, and Lü, (2006)” the reference should be added to the References list and cited in the text with its consecutive number

Lines 370-371 “to synthesis compositionally control the TixSiy oxide particles,” change to: “for a compositionally controlled synthesis of the TixSiy oxide particles,”

Lin 689 after “respectively” add “as reported in Table 6”

Line 741 “degradation of MB of PLA” change to: degradation of MB on PLA”

Author Response

Response to Reviewer 1 Comments

The paper is interesting and well written; however, there are some issues, particularly regarding the interpretation of the FTIR spectra.

A list of the main corrections request is listed below:

-          Lines 15-17 “The effect of pH 15 and the Ti/Si atomic ratio of titanium-silicon binary oxide (TixSiy) on the structural characteristics of TixSiy oxide are investigated by using the sol-gel technique”

Actually the sol-gel technique is used to prepare the samples, not to investigate their properties; the sentence should be rephrased.

The sentence has already been rewritten. (line 15-18)

Titania-silica oxides (TixSiy oxides) were successfully prepared via sol-gel technique. The Ti and Si precursors were titanium (IV) isopropoxide (TTIP) and tetraethylorthosilicate (TEOS), respectively. In this work, the effect of pH and the Ti/Si atomic ratio of titanium-silicon binary oxide (TixSiy) on the structural characteristics of TixSiy oxide are reported.

-          The TTIP/TEOS ratios for sample Ti50Si50 prepared at pH=9, presented Table 1 and 2, should be the same but they are not. The authors should explain. 

Sorry for the error from the data input. The data are now changed to the correct value.

-          In Figure 6 and 14, the FTIR spectra of TiO2 and SIO2 look very much alike. In the discussion of the FTIR spectra of the Ti50Si50 samples (lines 307-322), the band at 1052 cm-1 is attributed to asymmetric stretching vibrations of Si-O-Si of silica, but this band is also present in the spectrum of TiO2. On the other hand, the authors mention a Ti-O-Ti band that should be located at 1400 cm-1, but this band is absent in the spectrum of TiO2. I think this part of the paper should be checked and rewritten.

In particular, the sentence “the absence of the Ti-O-Ti bond in the oxide samples may be too weak to be detected at 1400 cm-1” (lines 319-320) should be rewritten.

The sentence has already be rewritten as the reviewer’s comments. (line 353-355 and line 365-374)

. TiO2 and all Ti50Si50 oxides displayed a prominent band at 400 to 700 cm-1 in their FTIR spectra, which corresponded to bending and stretching mode of Ti–O–Ti and characteristic of well-ordered TiO6 octahedrons.

The strong band near 3440 and 1630 cm-1 appeared in Figure 6, which are assigned to the streching and deformation vibration of the hydrocyl groups present (TiO2-OH) in the surface of TiO2 and all mixed oxide samples. The intensive band of OH-group asymmetrical and symmetrical stretching vibrations at 3440 cm-1 and O-H deformation vibration at 1630 cm-1 could confirm the large amount of water molecules. However, the OH-group vibration bands were substantially weaker as the calcination temperatures increased. The peaks at 440, 801 and 1052 cm−1 corresponded to the rocking, symmetric, and asymmetric stretching vibrations of Si-O-Si of silica, respectively. However, TiO2 also presents the peak at 1000-1200 cm-1, which is assigned to the deformation vibration of Ti-O-Ti of TiO2.

-          In the SEM images (figure 8 and 15), the enlargement used are shown in the picture, but they are almost impossible to read; the authors could indicate the enlargement used in the figure caption also.

The authors have indicated the enlargement detail in the figure caption already

The correction has already been made according to the reviewer’s comments in the figure caption (figure 8 and 15). (Line 408-409 and 683-684)

-          In the discussion of the XRD results (lines 373-384) the XRD patterns of sample Ti50Si50pH10 reveal the formation of anatase. I have one question to the authors: shouldn’t anatase formation be detectable from FTIR spectra too?

Thank you very much for your comment. We already add more information according to your comment as follow: (line 353-355)

TiO2 and all Ti50Si50 oxides displayed a prominent band at 400 to 700 cm-1 in their FTIR spectra, which corresponded to bending and stretching mode of Ti–O–Ti and characteristic of well-ordered TiO6 octahedrons.

-          In the discussion of the EXAFS result (lines 450-465 and table 3), the Ti-O bond distance for sample Ti50Si50pH8 (1.99 Å) does not seem to agree with the value expected for ti-O bond distance in a tetrahedral environment (1.83 Å) and closer to the value expected for anatase (1.95 Å); the authors should comment on that.

The comment on this issue has been added according to your comment.

The evidence of a longer Ti-O distance at 1.99 at pH 8.0 indicates a six-fold Ti site (octahedral) agrees with Loshmanov et al. who investigated TiO2 in SiO2 with neutron diffraction and attempted to isolate the Ti-O bond by subtracting their radial distribution functions from that of pure SiO2. They assigned a value of 1.95 A. By analogy with the Ti-O distance in its octahedrally-bonded oxides (1.91-2.01Å) they inferred an octahedral coordination for Ti in SiO2. This may be due to the effect of pH on the reaction of TixSiy oxides synthesized. (line 517-523)

-          Concerning the determination of the particle diameter (lines 504-513), I guess SD (line 509) stands for standard deviation and N=Sini; what does PDI in table 5 stand for?

The meaning of PDI was added according to your comment as follows:

Polydispersity index (PDI) is used to describe the degree of “non-uniformity” of a distribution. (Table 5)

-          Line 628-630 “However, the pre-edge peaks for TixSiy oxides are not as powerful as those predicted for Ti in a completely tetrahedral environment.” A citation is needed.

A citation was added according to your comment.

 The features and position in the rising edge of the B shoulders are clearer and the positions are lower. This suggest that the Ti-O-Ti linkages in TixSiy oxides increase and are incomplete anatase [1]. (line 705-707)

All Other minor corrections are made according to your comments as listed below:

Line 15 “Titania-silica oxides (TixSiy oxides) was prepared” change to “Titania-silica oxides (TixSiy oxides) were prepared”

New sentence:   Titania-silica oxides (TixSiy oxides) were successfully prepared via sol-gel technique.  (line 15)

Line 25 “ resulting in a respectable nanostructure.” Respectable is a strange adjective; I suggest using another one.

New sentence:   “resulting in an acceptable nanostructure.” (line 27)

Line 27 “a significant pre-edge feature in Ti50Si50 oxide samples” ; change to “a significant pre-edge feature in the spectra of Ti50Si50 oxide samples” 

New sentence:   “a significant pre-edge feature in the spectra of Ti50Si50 oxide samples “ (Line 29-30)

Line 33”degrading MB.” Add: (methylene blue)

New sentence:    “in degrading methylene blue (MB).”(line 35-36)

Line 34 “PLA/ i70Si30” change to:  “PLA/ Ti70Si30

New sentence:    “PLA/ Ti70Si30 “ (line 37)

Line 48 “Titanium dioxide” change to: “titanium dioxide”

New sentence:   “ titanium dioxide” (line 51)

Line 78 “factors to determining” change to: “factors determining”

New sentence:    “factors determining” (line 106)

Line 210 “200 mL of MB solution (10 mg/L) was added” change to:  “200 mL of MB solution (10 mg/L) were  added”

New sentence:    “200 mL of MB solution (10 mg/L) were added.” (line 255-256)

Line 291 “was reported by Pabón, Retuert, and Quijada (2007).” I guess it is reference [17] and so it should be cited in the text 8 if it is not, it should be added to the References list”

New sentence:    Similar observation was reported by Pabón, Retuert, and Quijada (2007) [2]. (line 336)

 Line 294 “bonds was created” change to “bonds were created”

New sentence:   “bonds were created”(line 339)

Line 368 “Wu, Wu, and Lü, (2006)” the reference should be added to the References list and cited in the text with its consecutive number

New sentence:   Wu, Wu, and Lü, (2006) claimed that the colloidal sol–gel technique can generate large particles composed of agglomerated nanoparticles and either condensed or porous polycrystalline microparticles [3]. (line 421-424)

Refference:

  1. Wu, L.; Wu, Y.; Lü Y. Self-assembly of small ZnO nanoparticles toward flake-like single crystals, Materials Research Bulletin 2006, 41, 128-133.

Lines 370-371 “to synthesis compositionally control the TixSiy oxide particles,” change to: “for a compositionally controlled synthesis of the TixSiy oxide particles,”

New sentence:   “to a compositionally controlled synthesis of the TixSiy oxide particles” (line 424-425)

Lin 689 after “respectively” add “as reported in Table 6”

New sentence:    “respectively as reported in Table 6.” (line 763)

Line 741 “degradation of MB of PLA” change to: “degradation of MB on PLA”

 New sentence:   “degradation of MB on PLA” (line 818)

Reviewer 2 Report

In this manuscript, the authors prepared TixSiy oxides with different pH condition and Ti/Si atomic ratio using sol-gel technique. The structural characteristics and the photocatalytic behavior of the TixSiy oxide samples were then investigated. I would suggest the acceptance of the manuscript after the following revision.

1. The title of the manuscript highlights the structural characterization of TixSiy oxides using Synchrotron X-ray absorption spectroscopy. However, the use of the synchrotron technique is not mentioned in the abstract or in the introduction part. I suggest the authors to revise the title of the manuscript or include some discussion about the novelty of using synchrotron X-ray absorption spectroscopy in the manuscript.

2. Figure 9, the authors mentioned that "no rutile phase was detected in any case". I would think the synthesized TiO2 powder present both the anatase and rutile phases. It's better to label the peaks in the spectra for clarification. In addition, have the authors calculated the crystallite size using Scherrer equation and compared the results with that measured in SEM?

3. For the FTIR spectra, have the authors compared the peak intensity differences in different samples by normalizing the peaks?

4. Figure 15, the scale bar is too small to read. It looks like the size of the particle in Figure 15(d) is smaller than those in Figure (b) and (c). But results in Table 5 show that the particle size of Ti40Si60 is the largest. 

5. Some papers about the structural characterization of particles using synchrotron technique can be cited: "High pressure induced atomic and mesoscale phase behaviors of one-dimensional TiO2 anatase nanocrystals." MRS Bulletin (2022): 1-6.; "Shape dependence of pressure-induced phase transition in CdS semiconductor nanocrystals." Journal of the American Chemical Society 142.14 (2020): 6505-6510.; "Structural characterizations of sol–gel synthesized TiO2 and Ce/TiO2 nanostructures." Physica B: Condensed Matter 407.15 (2012): 2915-2918.; Structural characteristics and reactivity/reducibility properties of dispersed and bilayered V2O5/TiO2/SiO2 catalysts." The Journal of Physical Chemistry B 103.4 (1999): 618-629.

Author Response

Response to Reviewer 2 Comments

In this manuscript, the authors prepared TixSiy oxides with different pH condition and Ti/Si atomic ratio using sol-gel technique. The structural characteristics and the photocatalytic behavior of the TixSiy oxide samples were then investigated. I would suggest the acceptance of the manuscript after the following revision.

  1. The title of the manuscript highlights the structural characterization of TixSiy oxides using Synchrotron X-ray absorption spectroscopy. However, the use of the synchrotron technique is not mentioned in the abstract or in the introduction part. I suggest the authors to revise the title of the manuscript or include some discussion about the novelty of using synchrotron X-ray absorption spectroscopy in the manuscript.

Thank you for you suggestion. An additional information was added according to your suggestion in Abstract and Introduction part of the manuscript.(line 21-23 , line 33, line 76-100, and line 111-113)

“By applying X-ray absorption spectroscopy (XAS) obtained from Synchrotron light sources, the qualitative characterization of the Ti-O-Si and Ti-O-Ti bonds in Ti-Si oxides was proposed.” (line 21-23)

“using Extended X-ray absorption fine structure (EXAFS) analysis” (line 33)

“The powerful technique of synchrotron X-ray absorption spectroscopy (XAS) using tuneable, very intense x-rays from a high energy electron storage ring has been applied to investigate the structural properties of materials. XAS is particularly attractive because of its ability to deliver electronic structure as well as geometric information. A typical K-edge absorption spectra is divided into two sections: (i) X-ray Absorption Near-Edge Structure (XANES) (<50 eV) contains bond lengths and angles, as well as information about the three-dimensional structure around the absorbing atom. (ii) Extended X-ray Absorption Fine Structure (EXAFS) region (typically > 50 eV) provides information on the initial coordination shell around the absorbing atom, including coordination numbers and bond lengths [18].

Most of the previous reports, XAS has been employed to determine information on the coordination environment of tetravalent Ti[Ti(IV)] in structurally complex of all oxide materials and bond distances between Ti-O and Ti-Si atoms. Niltharach et al. [19] used XANES methods to investigate the structural details of the sol–gel produced TiO2 samples with and without the inclusion of Ce. The XANES results also showed that the sample produced under the low hydrolysis condition had a significant number of Ti atoms in forms other than anatase and rutile TiO2. Won Bae Kim et al. [20] used the linear combination of two reference XANES spectra to estimate the pre-edge of the Ti K-edge in order to quantitatively analyze the percentages of Ti-O-Si and Ti-O-Ti bonds. The findings of pre-edge fitting in conjunction with XRD and XPS suggested that monolayer coverage was attained at around 7–10 wt.% Ti loading, where the concentration of Ti in Ti-O-Si was saturated to 0.56 mmol-Ti/g material. Shuji Matsuo et al.[21] determined the local Ti environments in the sol, gel, and xerogels of titanium oxide prepared by a sol-gel method using Titanium K-edge XANES. All of the samples may be divided into three groups: the anatase group, the anatase-like structure group, and the weak Ti–Ti interaction group.” (line 76-100)

“XANES and EXAFS were used to evaluate Ti-O-Si and Ti-O-Ti connectivity in TixSiy oxides, the local atomic structure, bond distances between Ti-O and Ti-Si atoms, coordination number, and valence state of titanium atoms TiO2 and all mixed oxide samples.” (line 111-114)

  1. Figure 9, the authors mentioned that "no rutile phase was detected in any case". I would think the synthesized TiO2 powder present both the anatase and rutile phases. It's better to label the peaks in the spectra for clarification. In addition, have the authors calculated the crystallite size using Scherrer equation and compared the results with that measured in SEM?

Thank you for your excellent guidance. The peaks in the spectra have already been labeled by the writers for clarity. (Figure 9 and 16)

The authors do not employ the Scherrer equation to compute the crystallite size. Then, particle size was only determined via SEM and diffraction particle size analysis.

  1. For the FTIR spectra, have the authors compared the peak intensity differences in different samples by normalizing the peaks?

No, we have not compared the peak intensity. We just employed FTIR for quality analysis in this study.

  1. Figure 15, the scale bar is too small to read. It looks like the size of the particle in Figure 15(d) is smaller than those in Figure (b) and (c). But results in Table 5 show that the particle size of Ti40Si60 is the largest. 

The correction has already been made according to the reviewer’s comments in the figure caption (figure 8 and 15). And the size of the scale bar has been rescaled by the writers.

  1. Some papers about the structural characterization of particles using synchrotron technique can be cited: "High pressure induced atomic and mesoscale phase behaviors of one-dimensional TiO2 anatase nanocrystals." MRS Bulletin(2022): 1-6.; "Shape dependence of pressure-induced phase transition in CdS semiconductor nanocrystals." Journal of the American Chemical Society142.14 (2020): 6505-6510.; "Structural characterizations of sol–gel synthesized TiO2 and Ce/TiO2 nanostructures." Physica B: Condensed Matter 407.15 (2012): 2915-2918.; Structural characteristics and reactivity/reducibility properties of dispersed and bilayered V2O5/TiO2/SiO2 catalysts." The Journal of Physical Chemistry B 103.4 (1999): 618-629.

Thank you very much for your recommended paper. We've already cited it and supplemented it with additional information based on your suggestion.

Submission Date

08 June 2022

Date of this review

11 Jun 2022 23:17:59

Round 2

Reviewer 1 Report

The paper has been modified as requested and is substantially improved. However, there is a problem with the paper's layout, at least in the pdf file I downloaded.

Figure 1 overlaps with table 1.

Figures 4,5,12,13,,14,16,19 and 20 are cut out.

Please, do this last minor corrections, before the paper is published.

Author Response

We have fixed the problem with the paper's layout and also extensive editing of English language was done by MDPI English editing service.
